# FLICKERFUSION: INTRA-TRAJECTORY DOMAIN GENERALIZING MULTI-AGENT RL

**Woosung Koh**[1]***Wonbeen Oh**[1], **Siyeol Kim**[1], **Suhin Shin**[1], **Hyeongjin Kim**[1], **Jaein Jang**[1],
**Junghyun Lee**[2], **Se-Young Yun**[2]†
[1]Yonsei University, `reiss.koh@yonsei.ac.kr`
[2]KAIST AI, `{jh_lee00, yunseyoung}@kaist.ac.kr`

## ABSTRACT

Multi-agent reinforcement learning has demonstrated significant potential in addressing complex cooperative tasks across various real-world applications. However, existing MARL approaches often rely on the restrictive assumption that the number of entities (e.g., agents, obstacles) remains constant between training and inference. This overlooks scenarios where entities are dynamically removed or *added during* the inference trajectory—a common occurrence in real-world environments like search and rescue missions and dynamic combat situations. In this paper, we tackle the challenge of intra-trajectory dynamic entity composition under zero-shot out-of-domain (OOD) generalization, where such dynamic changes cannot be anticipated beforehand. Our empirical studies reveal that existing MARL methods suffer *significant* performance degradation and increased uncertainty in these scenarios. In response, we propose FLICKERFUSION, a novel OOD generalization method that acts as a *universally* applicable augmentation technique for MARL backbone methods. FLICKERFUSION stochastically drops out parts of the observation space, emulating being in-domain when inferenced OOD. The results show that FLICKERFUSION not only achieves superior inference rewards but also *uniquely* reduces uncertainty vis-à-vis the backbone, compared to existing methods. Benchmarks, implementations, and model weights are organized and open-sourced at **`flickerfusion305.github.io`**, accompanied by ample demo video renderings.

## 1 INTRODUCTION

Multi-agent reinforcement learning (MARL) is gaining significant attention in the research community as it addresses real-world cooperative problems such as autonomous driving (Antonio & Maria-Dolores, 2022), power grid control (Biagioni et al., 2022), and AutoML (Wang et al., 2023b). Nevertheless, MARL research is commonly *over-simplified*, in that often, restrictive assumptions are made. Among these assumptions, we focus on one where the number of entities (e.g., agents, obstacles) available during training and inference is the *same* (Wang et al., 2021b; Liu et al., 2023; Zang et al., 2023; Hu et al., 2024). In response, Iqbal et al. (2021) introduced the concept of *dynamic team composition*, where the number of agents varies between training and testing. This concept has been further explored in subsequent works (Hu et al., 2021; Wang et al., 2023a; Shao et al., 2023; Tian et al., 2023; Yuan et al., 2024). Other variations including situations where the policies of uncontrollable teammates arbitrarily change intra-trajectory have also been studied (Zhang et al., 2023b). Despite extensive research, two crucial characteristics ((**1**) and (**2**)) frequently encountered in real-world MARL deployments have been overlooked.

(**1**) **Intra-trajectory Entrance.** Consider the following examples where entities *enter and leave intra-trajectory*. During a post-tsunami search and rescue (SAR) mission, additional obstacles may enter and leave *during* the inference trajectory (Drew, 2021), potentially due to secondary hazards like collapsing infrastructure. Similarly, in national defense, additional adversaries or allies dynamically enter and leave *during* the inference trajectory (Asher et al., 2023). Whether they are allies that

---

*Work done while an intern at KAIST AI

†Corresponding author

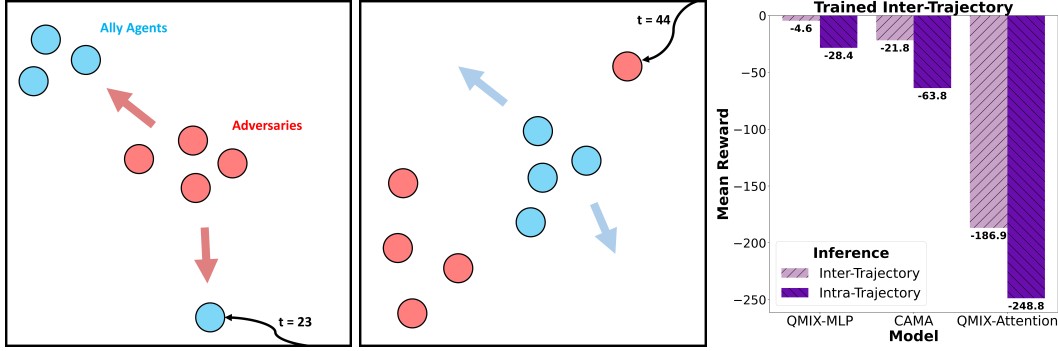

(a) Intra-trajectory entity addition at $t = 23$ of an ally agent (b) Intra-trajectory entity addition at $t = 44$ of an adversary (c) Train (3M steps) and inference with equal number of entities.

Figure 1: Motivating visualizations for intra-trajectory domain generalizing dynamic entity composition in the Tag environment. The models are trained on varying numbers of preys and predators sampled uniformly between 1 and 4. During inference (test-time), each scenario starts with one prey and one predator, with an additional 1 to 3 entities added *during* the trajectory.

require cooperation or adversaries and obstacles that the allies need to adapt to, this requires on-the-fly adaptation. Exiting entities have been studied in MARL, such as SMAC (Samvelyan et al., 2019) where units die, but scenarios where entities enter intra-trajectory have not yet been studied.

Despite seemingly small differences, the dynamics of entering entities differ from dying entities. Consider a simple predator-prey environment, Tag, in which the ally agents must move to avoid being tagged by adversaries. Unlike the simple case where an entity dies, intra-trajectory introduces novel scenarios where new entities enter from the outskirts of the observed region. For instance, when an ally enters, the existing allies and the new ally can enact a dispersion strategy to spread the attention of adversaries (Fig. 1a). Or, when a new adversary is introduced, the allies must immediately shift their initially planned trajectory to evade the new adversary (Fig. 1b). The agents' decentralized policies must adapt to this sudden shift in the scenario, *without* additional training intra-trajectory. Indeed, as shown in Fig. 1c, existing MARL approaches' performance deteriorates when entities enter intra-trajectory.

**(2) Zero-shot Domain Generalization.** Such performance degradation is more pronounced when the agents are presented with an entity composition they have never been explicitly trained on. For instance, in Tag, even though agents are trained with scenarios with at most four adversaries, they may be presented with five or more adversaries during inference. One naïve solution is to explicitly train on *every* possible dynamic scenario *a priori*, and this is the approach taken by prior literature on dynamic team composition (Iqbal et al., 2021; Hu et al., 2021). The key underlying assumption is that one knows the number of entities *beforehand* during training, which is unrealistic. In search and rescue (SAR) missions (Niroui et al., 2019), one cannot know all possible combinations of obstacles and targets (humans to rescue) in advance; in national defense, knowing all potential adversaries' combinations during combat *a priori* is impossible. Even when some information of the inference combination is available *a priori*, the total number of combinations can be very large and impractical to train on. This motivates us to study these dynamic MARL scenarios under zero-shot out-of-domain (OOD) generalization (Min et al., 2020), where here, domain refers to the entity composition.

**Prior Attempts.** These unexpected dynamic scenarios present a non-trivial problem—each agent is forced to observe an input-space *greater* than it was trained on. Prior works (Hu et al., 2021; Zhang et al., 2023a; Shao et al., 2023) all took the approach of appropriately expanding the $Q$ or policy network, via **additional parameters**, using inductive bias. However, empirically, all fail to deliver reasonable performance in the considered OOD MARL environments, let alone being consistent, across environments, and within environments (seeds)—even after extensive hyperparameter tuning (Table 1; Fig. 5 (top-left)). The high uncertainty of existing methods especially renders them unreliable for safety-critical applications (Knight, 2002) such as autonomous vehicles (Pek et al., 2020; Feng et al., 2023), where the deployed method must be robust to changes between train and test-time.

**Contribution.** We propose a novel, *orthogonal* approach for OOD MARL, FLICKERFUSION, that augments the agents' observation space, introducing **no additional parameters** test-time. The augmentation is based on the widely-adopted dropout mechanism (Srivastava et al., 2014)—where neurons (including the input layer), are randomly masked out. FLICKERFUSION differs from Srivastava et al. (2014) as it leverages the dropout mechanism only in the input layer, such that, the model can remain in an emulated in-domain regime, even when inferenced OOD. A high-level schematic overview is provided in Fig. 2. Prior works do not take this alternative approach as an immediate concern becomes, given no additional parameters, parts of the input space are lost.

The stochasticity of the augmentation effectively preserves the information of the entire observation—aggregating the stochastic partial observations across time results in an emulated full observation. Note that the *maximum number of entities need not be known beforehand*, as the entity dropout can adapt to any OOD entity composition at test-time. The training is done similarly to "prepare" for the entity dropouts that the agents will experience at test time (**Sec. 3**). Remarkably, this simple **lose** and **recover** approach attains state-of-the-art performance vis-à-vis *inference reward* and *uncertainty reduction*, relative to an *extensive* list of existing MARL and other model-agnostic methods (**Sec. 5**). Finally, as no prior benchmarks can evaluate such dynamic MARL scenarios, we also standardize and present 12 benchmarks (**Sec. 4**).

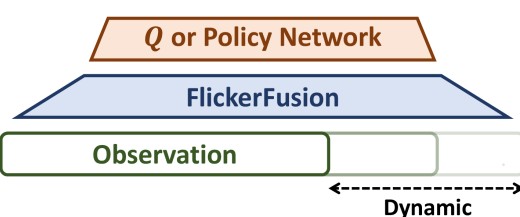

Figure 2: Schematic diagram of FLICKERFUSION. FLICKERFUSION can be *universally* attached between *any* dynamic observation space and *any* $Q$ or policy network. It ensures the network does not have to initialize extra parameters even as observation size changes at test-time.

**Analogy with the Flicker Fusion Phenomena.** FLICKERFUSION is named after the infamous Flicker Fusion phenomena in physiology (Simonson & Brozek, 1952). The Flicker Fusion phenomena occurs when an agent perceives a flickering light source as a steady, non-flickering illusion as the frequency of the flicker passes a certain threshold (Landis, 1954; Levinson, 1968). Here, dropping entities from agents' observations represents *flickers*, and through a stochastic input-space drop out at each time step, we *fuse* the flickers to emulate a full-view across the temporal dimension. We revisit this analogy in **Sec. 3.3**. Video visualizations of FLICKERFUSION are available on our **project site**.

## 2 PRELIMINARIES

**Notations.** For a positive integer $n \in \mathbb{N}$, denote $[n] := \{1, 2, \cdots, n\}$. For two vectors $\boldsymbol{a}$ and $\boldsymbol{b}$, $\boldsymbol{a} \oplus \boldsymbol{b}$ is the concatenated vector $[\boldsymbol{a} \ \boldsymbol{b}]$. For a set $A$, $|A|$ is its cardinality, $2^A$ is its power set, $\Delta(A)$ is the probability distribution's possibility set over $A$, and $A^*$ is the set of all possible $A$-valued sequences of arbitrary lengths. Also, for an integer $n \leq |A|$, $\binom{A}{n} := \{B \subset A : |B| = n\}$. For a $m \times n$ matrix $\boldsymbol{A}$, denote $\text{nrow}(\boldsymbol{A}) = n$ to be the number of row.

**Dec-MDP with Dynamic Entities.** We follow the *de facto* MARL framework, Centralized Training Decentralized Execution (CTDE; Rashid et al. (2018); Foerster et al. (2017)). Our setting can be described as a Decentralized Markov Decision Process (Dec-MDP; Bernstein et al. (2002); Oliehoek & Amato (2016)) with *dynamic* entities (Iqbal et al., 2021; Schroeder de Witt et al., 2019), described by the tuple $\langle \mathcal{E} := \mathcal{E}_a \dot\cup \mathcal{E}_n, \boldsymbol{\Phi}, \phi, \boldsymbol{S}, \boldsymbol{U}, P, \widetilde{P}, R, \gamma \rangle$. $\mathcal{E}$ is the set of all possible entities, that may enter or leave intra-trajectory, and $\mathcal{E}$ is partitioned into the set of (trainable) agents $\mathcal{E}_a$ and other non-agent entities $\mathcal{E}_n$. For a $L \in \mathbb{N}$, we assume each entity takes one of $L$ types (e.g., ally, adversary, obstacle), and $\boldsymbol{\Phi} := \{\boldsymbol{e}_1, \boldsymbol{e}_2, \cdots, \boldsymbol{e}_L\}$ is the set of all possible entity types where $\boldsymbol{e}_\ell$'s are the elementary bases of $\mathbb{R}^L$. For each entity $e \in \mathcal{E}$, we denote $\phi^e \in \boldsymbol{\Phi}$ as its type.

**State-Action Spaces.** Let $\mathcal{S} \subseteq \mathbb{R}^{d_s}$ be the shared state space of all the entities, and assume that each entity $e \in \mathcal{E}$ has a state representation $s^e \in \mathcal{S}$. Following Iqbal et al. (2021), we assume that $s^e := f^e \oplus \phi^e$ where $f^e \in \mathbb{R}^{d_{\phi^e}}$ is the feature vector (e.g., velocity, location). To deal with potentially entering and leaving of entities, we define the joint state space $\boldsymbol{S}$ as

the space of *joint states of all possible entity combinations*: $\boldsymbol{S} := \bigcup_{E \subseteq \mathcal{E}} \boldsymbol{S}(E)$ with $\boldsymbol{S}(E) := \left\{ \boldsymbol{s}^E \triangleq \{(e, s^e) : e \in E\} : s^e \in \mathcal{S} \right\}$ being the space of all possible joint states for a given entity combination $E \subseteq \mathcal{E}$. We assume full observability, i.e., each agent $a$ at their current state $\boldsymbol{s}^a$ can observe $\boldsymbol{o}^a := \{(e, s^e) : e \in E\} \in \boldsymbol{S}$. Similarly, the joint action space $\boldsymbol{U}$ is defined as $\boldsymbol{U} := \bigcup_{E_a \subseteq \mathcal{E}_a} \boldsymbol{U}(E_a)$ with $\boldsymbol{U}(E_a) := \left\{ \boldsymbol{u}^{E_a} \triangleq \{(a, u^a) : a \in E_a\} : u^a \in \mathcal{U} \right\}$ for some shared action space $\mathcal{U}$.

**Policy Learning.** Under decentralized execution, each agent $a$ has its own policy $\pi^a : \tau^a \mapsto \pi^a(\cdot|\tau^a) \in \Delta(\mathcal{S})$, where $\tau^a \in (\boldsymbol{S} \times \boldsymbol{U})^*$ is the observation-action trajectory available to agent $a$. Then, the joint policy $\boldsymbol{\pi}$ is defined as $\boldsymbol{\pi} : (E_a, (\tau^a)_{a \in E_a}) \mapsto (\pi^a(\cdot|\tau^a))_{a \in E_a}$, where $E_a \subseteq \mathcal{E}_a$ is the current agent composition. The global reward function is $R : \boldsymbol{S} \times \boldsymbol{U} \mapsto \mathbb{R}$, which we assume to be deterministic. The global state-action Q-value is defined as

$$Q^{\boldsymbol{\pi}}(\boldsymbol{s}, \boldsymbol{u}, E) := \mathbb{E}\left[ \sum_{t=0}^{\infty} \gamma^t R(\boldsymbol{s}_t, \boldsymbol{u}_t) \middle| \boldsymbol{s}_0 = \boldsymbol{s}, \boldsymbol{u}_0 = \boldsymbol{u}, E_0 = E \right], \tag{1}$$

where $\gamma \in (0, 1)$ denotes the discount factor, and $\mathbb{E}$ is taken *w.r.t.* the randomness of $P$, $\widetilde{P}$, and $\boldsymbol{\pi}$. Eq. (1) is trained centrally via QMIX (Rashid et al., 2018):

$$Q_{tot}(\boldsymbol{\tau}, \boldsymbol{u}; \boldsymbol{\theta}) := g\left( Q^1(\tau^1, u^1; \boldsymbol{\theta}_Q^1), \cdots, Q^N(\tau^N, u^N; \boldsymbol{\theta}_Q^N); h(\boldsymbol{s}, \boldsymbol{\theta}_h) \right), \tag{2}$$

where $g(\cdot, \cdot; h(\boldsymbol{s}, \boldsymbol{\theta}_h))$ is a monotonic mixing function, parameterized by a hypernetwork (Ha et al., 2017) conditioned on the global state $\boldsymbol{s}$. The parameters $\boldsymbol{\theta} = \{\boldsymbol{\theta}_Q^1, \cdots, \boldsymbol{\theta}_Q^N, \boldsymbol{\theta}_h\}$ are centrally trained with the DQN loss (Mnih et al., 2013). After training, the inference for each agent $a$ is done decentrally as choosing $\arg\max_u Q^a(\tau^a, u; \boldsymbol{\theta}_Q^a)$.

## 3 FLICKERFUSION: A NEW APPROACH TO OOD MARL

As mentioned in the Introduction, we take a different direction from existing methods. Therefore, we first revisit the two most popular MARL backbones and unpack the innate cause of domain-shift performance degradation under them (**Sec. 3.1**), then introduce our approach (**Sec. 3.2, 3.3**).

### 3.1 TOWARDS DOMAIN GENERALIZATION ACROSS ENTITY COMPOSITION

In our context, **domain** is a set of tuples $\{(N_\ell^l, N_\ell^u)\}_{\ell \in [L]} \subset \mathbb{N}_0^L$ such that

$$N_\ell^l \leq \min_{t \geq 0} |\{e \in E_t : \phi^e = \boldsymbol{e}_\ell\}| \leq \max_{t \geq 0} |\{e \in E_t : \phi^e = \boldsymbol{e}_\ell\}| \leq N_\ell^u \ a.s., \quad \forall \ell \in [L]. \tag{3}$$

In other words, the number of type $\boldsymbol{e}_\ell$ entities is between a lower bound, $N_\ell^l$, and an upper bound, $N_\ell^u$. Given training occurs in-domain, $\mathcal{D}_I$, then, it is **out-of-domain (OOD)**, $\mathcal{D}_O$, if there exists at least one $\ell \in [L]$ such that $N_\ell^l$ of $\mathcal{D}_O$ is strictly greater than $N_\ell^u$ of $\mathcal{D}_I$ (see "Out-of-Domain" in Fig. 3a). Thus, in $\mathcal{D}_O$, the learner faces entity-type combinations that she has never seen before. We illustrate why a naïve application of usual MARL backbones (QMIX-MLP and QMIX-Attention) are sub-optimal for domain generalization. Note that while we focus on off-policy backbones as they are the most commonly used, all our discussion can be trivially extended to on-policy networks. Let us denote $\boldsymbol{\theta}_Q^{\mathcal{D}_I}$ as the parameters of the in-domain trained QMIX.

**MLP Backbone.** First, consider QMIX-MLP (Rashid et al., 2018), where an MLP parametrizes the $Q$-function. When using QMIX-MLP in OOD scenarios, one must expand $\boldsymbol{\theta}_Q^{\mathcal{D}_I}$ to $\boldsymbol{\theta}_Q^{\mathcal{D}_O} := \boldsymbol{\theta}_Q^{\mathcal{D}_I} \oplus \boldsymbol{\theta}_Q'$ to take OOD observation $\boldsymbol{o}_{\mathcal{D}_O}^a$ as input. This is because $|\boldsymbol{o}_{\mathcal{D}_O}^a| > |\boldsymbol{o}_{\mathcal{D}_I}^a|$ due to the additionally introduced entities, where $\boldsymbol{o}_{\mathcal{D}_I}^a$ is the in-domain observation. But, because no additional training is allowed during execution, the choice of initialization of $\boldsymbol{\theta}_Q'$ is critical.

Due to the structure of MLPs, $\boldsymbol{\theta}_Q'$ must be initialized without prior knowledge or inductive bias of the encountered OOD scenario, resulting in sub-optimal performance. This approach disregards one salient prior knowledge: the *entity type*, $\phi^e$. Suppose the newly added entity is the same type as the one the learner has seen during training. In that case, it is only natural to initialize the expanded part using corresponding parts of the already trained $Q$-network. This can be implemented using QMIX-Attention (Iqbal et al., 2021), with some essential modifications described below.

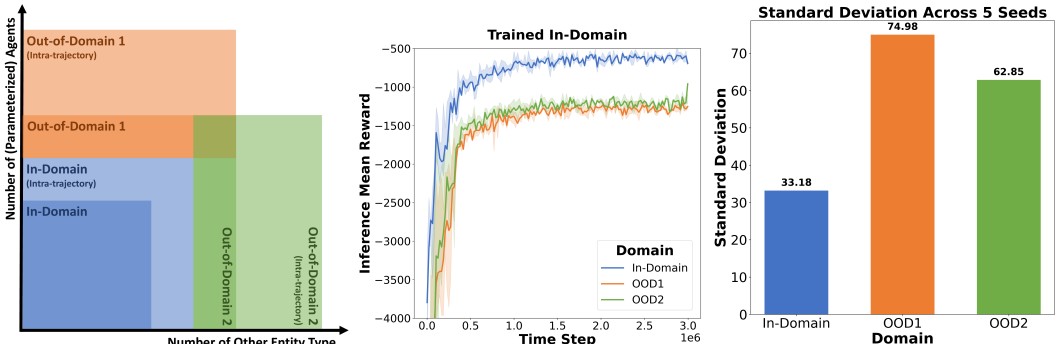

(a) Entity count domain generaliza-
tion (Orange: OOD1; Green: OOD2)

(b) Drop in reward under domain gen-
eralization

(c) Rise in uncertainty under domain
generalization

Figure 3: Example out-of-domain reward and uncertainty performance (color coded). Shaded region in (b) indicate quartile 1 to 3.

**Attention Backbone.** We consider the tokenizer (Vaswani et al., 2017) defined as $T(f^e, \phi^e) := MLP(f^e \oplus \phi^e)$, and let $\boldsymbol{X} \in \mathbb{R}^{N \times d_T}$ be a matrix whose each row corresponds to the tokenized embedding of each entity, and $d_T$ is the token embedding size. Then, even though we need to introduce additional parameter $\boldsymbol{\theta}'_Q$ to account for observation of higher dimensionality, it can be initialized by utilizing the tokenizer and the currently observed entity type. However, there is still another source of sub-optimality from the inherent structure of the attention mechanism and the fact that the size of the attention matrix changes, namely, $\mathrm{nrow}(\mathbf{X}^{\mathcal{D}_O}) > \mathrm{nrow}(\mathbf{X}^{\mathcal{D}_I})$ in

$$\text{Attention}(\mathbf{Q}, \mathbf{K}, \mathbf{V}) := \text{softmax}\left(\frac{\mathbf{Q}\mathbf{K}^\top}{\sqrt{d_k}}\right)\mathbf{V}, \tag{4}$$

$$\mathbf{Q} := \mathbf{X}_a\mathbf{W}^Q, \mathbf{K} := \mathbf{X}\mathbf{W}^K, \mathbf{V} := \mathbf{X}\mathbf{W}^V, \quad \mathbf{X} := \bigoplus_{\mathcal{E}}\textbf{Token}_i, \mathbf{X}_a := \bigoplus_{\mathcal{E}_a}\textbf{Token}_i, \tag{5}$$

where $\bigoplus$ denotes concatenating the vectors row-wise and $\boldsymbol{W}^Q, \boldsymbol{W}^K, \boldsymbol{W}^V \in \mathbb{R}^{d_T \times d_k}$ are trainable weights. Thus, even with the entity type $\phi^e$ prior, the change in $\mathbf{X}$'s size leads to sub-optimality in Eq. (4) and (5) as the model has never been trained on entity compositions larger than $\mathrm{nrow}(\mathbf{X}^{\mathcal{D}_I})$.

**Empirical Example.** For a concrete example, we visualize OOD performance degregation on one of the benchmarks we describe later in Sec. 4. As seen in Fig. 3b and 3c, both inference reward and uncertainty sharply deteriorate under OOD generalization. Although this example study uses a representative backbone MARL method, QMIX-Attention, this pattern of deterioration persists across different environments *and* methods.

## 3.2 FLICKERING VIA ENTITY DROPOUT

While past works remedy OOD generalization via injecting inductive biases within $\boldsymbol{\theta}'_Q$, our orthogonal approach ensures that $\mathrm{nrow}(\mathbf{X}^{\mathcal{D}_O}) = \mathrm{nrow}(\mathbf{X}^{\mathcal{D}_I})$, no longer requiring $\boldsymbol{\theta}'_Q$. In turn, the key design question becomes *how many* and *which entities* to drop.

**How Many to Drop?** We consider the approach of artificially limiting the visibility as follows:

$$\widetilde{\boldsymbol{X}}^{\mathcal{D}_O} \leftarrow \mathbf{X}^{\mathcal{D}_O}[\mathcal{I}, :], \quad \mathcal{I} \in \binom{[\mathrm{nrow}(\boldsymbol{X}^{\mathcal{D}_O})]}{\mathrm{nrow}(\boldsymbol{X}^{\mathcal{D}_I})}, \tag{6}$$

where $\boldsymbol{X}^{\mathcal{D}_O}[\mathcal{I}, :]$ is the submatrix of $\boldsymbol{X}^{\mathcal{D}_O}$ consisting of the rows of indices in index set $\mathcal{I}$. This ensure that $\mathrm{nrow}(\widetilde{\boldsymbol{X}}^{\mathcal{D}_O}) = \mathrm{nrow}(\boldsymbol{X}^{\mathcal{D}_I})$. This approach has a trade-off: one needs not to introduce new parameters $\boldsymbol{\theta}'_Q$ as the observation size is retained, but one loses information regarding the entities indexed by $\mathcal{I}' := [\mathrm{nrow}(\boldsymbol{X}^{\mathcal{D}_O})] \setminus \mathcal{I}$. We refer to this solution (Eq. (6)) as **domain-aware entity dropout (DAED)**. It is domain-aware as it artificially augments the observation such that it is of

the same size as in-domain. Under the attention backbone, this corresponds to masking the tokens indexed by $\mathcal{I}'$, which has been effectively used in prior MARL literature (Iqbal et al., 2021; Shao et al., 2023). Under an MLP backbone, the non-EDed $s^e$'s are concatenated.

The choice of $\mathcal{I}$ dictates which and how much information is lost, making it a critical design choice. Let $\boldsymbol{N}^{train} := [N_1^{train}, \cdots, N_L^{train}]$ where $N_\ell^{train}$ is the upper bound on the number of type $\boldsymbol{e}_\ell$ entities encountered during training (Eq. (3)). Let $\boldsymbol{N}^{inf} := [N_1^{inf}, \cdots, N_L^{inf}]$ be the vector of the number of entities per type that is encountered during inference. We desire $|\mathcal{I}'|$ to be as small as possible to minimize the information loss while augmenting the observation size to that of the in-domain. This is done by dropping $\max(0, N_\ell^{inf} - N_\ell^{train})$ entities for each type $\boldsymbol{e}_\ell$.

**Which to Drop?**  Now that we have determined how many entities of each type to drop, we discuss which ones to drop. Following Iqbal et al. (2021), we use randomized drop-outs: given $\Delta := \boldsymbol{N}^{inf} - \boldsymbol{N}^{train}$, each agent independently and uniformly samples $\Delta[\ell]$ number of entities of type $\ell$ to drop. This simple solution is decentralized across the agents and, "on-average", results in a "dispersed" view of the entities for the agents. This is formalized in the following statement, whose proof is deferred to Appendix A:

**Proposition 3.1.** *Let $[N_A]$ and $[N_\ell]$ be the set of agents and entities of type $\ell$, respectively. Suppose each agent drops $\Delta_\ell$ entities of type $\ell$, at uniform and independently. Let $d(i)$ be the (random) number of agents that have dropped the entity $i \in [N_\ell]$. Then, we have the following in-expectation guarantee:*

$$\mathbb{E}\left[\sum_{i \in [N_\ell]} \left| \frac{d(i)}{N_A} - \frac{\Delta_\ell}{N_\ell} \right| \right] \leq \sqrt{\frac{\Delta_\ell(N_\ell - \Delta_\ell)}{N_A}}. \tag{7}$$

$\widehat{\boldsymbol{\mu}} := (d(i)/N_A)_{i \in [N_\ell]}$ is the empirical ratios of the number of agents that have dropped entity $i$, and $\boldsymbol{\mu} := (\Delta_\ell/N_\ell)_{i \in [N_\ell]}$ is the uniform ratio. Thus, even with a random sampling scheme, $\|\widehat{\boldsymbol{\mu}} - \boldsymbol{\mu}\|_1$ is small, i.e., the entities are dropped in a way that is minimally overlapping across agents. While this encourages a dispersed view across the agents, if the entities to be dropped remain fixed throughout the trajectory, there is still some permanently lost information from the perspective of *each* agent.

## 3.3 FUSING THE FLICKERS

We now demonstrate how we can **recover** some of the lost information, further improving the trade-off of Eq. (6) to our favor. For each (active) agent $a$, let $D_a \subset E_t$ be the set of dropped entities from the perspective of $a$. If $D_a$ remains fixed throughout the trajectory, then $a$ never sees $D_a$. We alleviate this problem by *stochastically* changing $D_a$ at each $t$. This also has the added benefit that any intra-trajectory dynamics changes can be immediately responded to on-the-fly. In essence, FLICKERFUSION *fuses* (stochastic sampling at each $t$) the *flickers* (ED). The pseudocodes for train and inference modes are presented in Alg. 1 and 2.

**Train.**  Alg. 1 shows the pseudocode for FLICKERFUSION in the training phase, taking the in-domain entity composition $\boldsymbol{N}^{train}$ and learning frequency hyperparameter $b$ as inputs. For each episode, the environment (entity composition, line 3) and the number of entities to be dropped ($\Delta_t^{train}$, line 6) are both randomized. Here, ED$(\cdot, \cdot, \cdot)$ takes the agent $a$, vector of entities to be dropped $\Delta_t$, and entity list $E_t$, and outputs a randomly dropped out observation $\boldsymbol{o}_a$, satisfying Eq. (6). This ensures that the network is well-trained over various in-domain combinations (line 3) as well as various partial observations (line 6), where Uniform$(\boldsymbol{a}, \boldsymbol{b})$ outputs a random vector whose $i$-th coordinate is uniformly sampled between $\boldsymbol{a}_i$ and $\boldsymbol{b}_i$. We apply this training procedure to *all* the baseline algorithms for our empirical study later.

**Inference.**  Alg. 2 is the pseudocode for FLICKERFUSION in the inference phase, again taking the in-domain entity composition $\boldsymbol{N}^{train}$ as input. At each time $t$, the number of entities to be dropped $\Delta_t^{inf}$ is computed from the perspective of agent $a$, where Observe$(t, a)$ outputs the entity combination vector observed by agent $a$ (line 6). We implement stochastically changing $D_a$ by calling ED at each $t$ such the random partial observation ($\boldsymbol{o}_a$), as well as the set of agents not dropped ($N_{t,a}$), are obtained from DAED (line 7), using which the next action is

chosen greedily (line 9). Importantly, all this is done *decentrally* across agents at each time $t$.

| **Algorithm 1:** FLICKERFUSION (Train) | **Algorithm 2:** FLICKERFUSION (Inference) |
|---|---|
| 1 **Input:** $N^{train}, b$; | 1 **Input:** $N^{train}$; |
| 2 **for each** *episode* **do** | 2 **for** $t = 1, 2, \cdots$ **do** |
| 3 $\quad$ $\hat{N}^{train} \leftarrow \text{Uniform}(\mathbf{1}, N^{train})$; | 3 $\quad$ **for** $a \in E_{t,a}$ *(decentrally)* **do** |
| 4 $\quad$ CreateEnvironment($\hat{N}^{train}$); | 4 $\quad\quad$ **if** $a$ **is** *newly introduced* **then** |
| 5 $\quad$ **for** $t = 1, 2, \cdots$ **do** | 5 $\quad\quad\quad$ Initialize $\tau^a \leftarrow \{\}$; |
| 6 $\quad\quad$ $\Delta_t^{train} \leftarrow \text{Uniform}(\mathbf{0}, \hat{N}^{train})$; | 6 $\quad\quad$ $\Delta_t^{inf} \leftarrow \text{Observe}(t, a) - N^{train}$; |
| 7 $\quad\quad$ $\mathbf{o}_a, \_ \leftarrow \text{ED}(a, \Delta_t^{train}, E_t)$ for each | 7 $\quad\quad$ $\mathbf{o}_a, N_{t,a} \leftarrow \text{ED}(a, \Delta_t^{inf}, E_t)$; |
| $\quad\quad$ agent $a \in E_{t,a}$; | 8 $\quad\quad$ $\tau^a \leftarrow \tau^a \cup \{\mathbf{o}_a\}$; |
| 8 $\quad\quad$ Add to central replay buffer; | 9 $\quad\quad$ Choose |
| 9 $\quad\quad$ **if** $t \% b == 0$ **then** | $\quad\quad\quad$ $u_a \leftarrow \arg\max_u Q^a(\tau^a, u; \boldsymbol{\theta}_Q^a \in \boldsymbol{\theta})$; |
| 10 $\quad\quad\quad$ Batch sample from replay buffer; | |
| 11 $\quad\quad\quad$ Train $\boldsymbol{\theta}$ via QMIX; | 10 $\quad\quad$ $\tau^a \leftarrow \tau^a \cup \{\{u_e\}_{e \in N_{t,a}}\}$; |

To concretely demonstrate the effect of calling DAED every $t$ (line 7), consider a toy scenario where $L = 1$, $\text{nrow}(\boldsymbol{X}^{\mathcal{D}_I}) = 4$, and $\text{nrow}(\boldsymbol{X}^{\mathcal{D}_O}) = 5$. As illustrated in Fig. 4, each agent's view stochastically changes at each $t$. As seen by the **green**, executing the algorithm at every $t$ ensures smooth transition to any intra-trajectory OOD dynamics. The important intuition is that aggregating these views along the temporal axis gives us a virtually full view; this is illustrated more clearly on our **project site**.

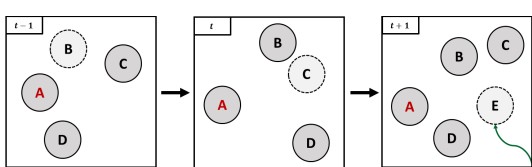

Figure 4: From the perspective of the **red** agent (A), the dash-lined and lighter background color entity is dropped out. Due to stochasticity, the entity dropped commonly differs across $t$.

# 4 MPEv2: AN ENHANCED MARL BENCHMARK

To ensure that this problem setting does not fall prey to the standardization issue raised by Papoudakis et al. (2021), we enhance Multi Particle Environments (MPE; Lowe et al. (2017)) to MPEv2, consisting of 12 open-sourced benchmarks. Although of a different nature, the enhancement is conceptually similar to the enhancements made to the StarCraft Multi-agent Challenge (SMAC; Samvelyan et al. (2019)) that led to SMACv2 (Ellis et al., 2023) with stochasticity and SMAC-Exp (Kim et al., 2023) with multi-stage tasks. While SMAC and MPE are the two most popular MARL environments (Rutherford et al., 2024), we choose to improve MPE due to its lower computational requirements (Papoudakis et al., 2021), ensuring accessible research. This is pertinent as OOD dynamic entity composition is inherently computationally burdensome, due to dimensionality explosion vis-à-vis the simulator (benchmark) and model (parameter) size.

MPEv2 consists of six environments, three extensions to the original MPE, and three novel. These environments were developed *a priori* to the experiments. All environments support an arbitrary dynamic entity composition, including intra-trajectory addition and deletion. As visualized in Fig. 3a, each environment has two benchmarks, OOD1 and OOD2, which assess generalization performance by increasing the number of parameterized agents and non-agent entities, respectively. Fine-grained details for each benchmark are presented in Appendix B.

**Brief Descriptions.** First, we take the three most appropriate (in terms of our dynamic scenarios) MPE environments, (1) **Spread**, (2) **Adversary**, (3) **Tag**, and add-on arbitrary dynamic entity scenarios. These are then standardized to six benchmarks, two for each environment. The remaining three environments are newly created. (4) **Guard** is a two-entity (agents, targets) environment where the agents act as bodyguards to the targets. The targets stochastically move point-to-point given a time interval. The agents' objective is to emulate guarding multiple moving targets. Furthermore, they are

Table 1: Empirical results, 3M steps (mean final inference reward $\pm\sigma$)

| MPEv2 Environment | Spread | | Repel | | Tag | |
|---|---|---|---|---|---|---|
| **Method** | OOD1 | OOD2 | OOD1 | OOD2 | OOD1 | OOD2 |
| FLICKERFUSION-Attention | -198.4 ±17.0 | -230.1 ±18.6 | 845.6 ±99.1 | 1070.2 ±215.4 | **-2.1 ±0.8** | **-9.4 ±3.8** |
| FLICKERFUSION-MLP | **-183.7 ±29.4** | -209.0 ±25.4 | 906.1 ±42.3 | **1189.0 ±48.6** | -18.8 ±24.3 | -33.7 ±15.9 |
| ACORM (Hu et al., 2024) | -244.4 ±24.1 | -330.3 ±23.6 | 153.6 ±773.2 | 98.1 ±726.6 | -247.9 ±174.1 | -578.2 ±763.7 |
| ODIS (Zhang et al., 2023a) | -257.1 ±38.2 | -376.5 ±84.6 | -3532.8 ±2994.9 | -2982.1 ±2074.4 | -1992.0 ±2331.4 | -1723.4 ±1817.3 |
| CAMA (Shao et al., 2023) | -262.2 ±100.6 | -464.1 ±244.4 | 698.2 ±126.2 | 928.1 ±198.9 | -29.6 ±37.8 | -31.2 ±28.4 |
| REFIL (Iqbal et al., 2021) | -205.0 ±41.0 | -289.2 ±29.2 | 404.6 ±351.2 | 822.4 ±148.4 | -9.3 ±13.0 | -15.2 ±9.8 |
| UPDeT (Hu et al., 2021) | -222.0 ±80.4 | -273.9 ±57.8 | -2645.2 ±5006.1 | -1929.0 ±3993.5 | -129.5 ±50.9 | -128.8 ±82.2 |
| Meta DotProd (Kedia et al., 2021) | -367.8 ±95.1 | -555.9 ±170.4 | -12538.6 ±526.5 | -9365.7 ±164.2 | -7611.4 ±7279.8 | -7302.5 ±7436.8 |
| DG-MAML (Wang et al., 2021a) | -217.3 ±17.9 | -254.0 ±15.3 | 343.2 ±438.0 | 623.0 ±420.4 | -39.2 ±27.5 | -50.8 ±30.8 |
| SMLDG (Li et al., 2020) | -258.2 ±38.4 | -290.4 ±32.8 | -1711.0 ±2093.7 | -1211.7 ±1868.4 | -210.6 ±120.5 | -185.9 ±148.8 |
| MLDG (Li et al., 2018) | -413.1 ±47.1 | -610.7 ±34.4 | -12763.4 ±213.5 | -9711.1 ±293.4 | -2625.9 ±4899.6 | -2333.2 ±4372.1 |
| QMIX-Attention (Iqbal et al., 2021) | -190.1 ±8.6 | -251.5 ±42.0 | 719.0 ±141.7 | 927.4 ±215.1 | -14.4 ±9.8 | -26.6 ±10.9 |
| QMIX-MLP (Rashid et al., 2018) | -209.1 ±27.4 | -242.8 ±37.3 | 380.0 ±195.0 | 883.7 ±225.6 | -41.6 ±16.5 | -62.5 ±36.4 |
| **MPEv2 Environment** | Guard | | Adversary | | Hunt | |
| **Method** | OOD1 | OOD2 | OOD1 | OOD2 | OOD1 | OOD2 |
| FLICKERFUSION-Attention | -1258.3 ±113.3 | -1160.0 ±62.6 | 60.9 ±24.5 | 9.9 ±25.9 | -297.1 ±20.2 | -337.5 ±12.9 |
| FLICKERFUSION-MLP | **-1127.2 ±66.2** | **-962.9 ±49.1** | 56.3 ±20.1 | 9.3 ±13.2 | **-278.5 ±17.2** | **-305.3 ±11.8** |
| ACORM (Hu et al., 2024) | -2074.3 ±323.8 | -2049.6 ±234.0 | 56.1 ±18.1 | 13.1 ±19.2 | -367.2 ±59.0 | -397.3 ±30.4 |
| ODIS (Zhang et al., 2023a) | -1449.3 ±80.1 | -1442.8 ±278.6 | -56.8 ±37.7 | -188.7 ±105.2 | -667.5 ±99.5 | -657.8 ±209.7 |
| CAMA (Shao et al., 2023) | -5002.5 ±2008.6 | -4656.9 ±2252.9 | 50.7 ±11.1 | 26.5 ±14.5 | -1063.1 ±397.3 | -1131.3 ±565.5 |
| REFIL (Iqbal et al., 2021) | -1445.8 ±196.8 | -1294.7 ±88.6 | 14.5 ±16.6 | -11.7 ±29.4 | -305.0 ±18.6 | -347.9 ±14.8 |
| UPDeT (Hu et al., 2021) | -2845.5 ±2294.0 | -2637.0 ±2393.4 | -64.6 ±80.1 | -0.6 ±59.6 | -340.8 ±82.7 | -383.5 ±110.5 |
| Meta DotProd (Kedia et al., 2021) | -7507.4 ±550.0 | -7620.4 ±504.2 | **82.3 ±15.5** | **34.4 ±20.8** | -956.0 ±518.1 | -1087.9 ±720.7 |
| DG-MAML (Wang et al., 2021a) | -1885.3 ±187.3 | -1989.1 ±288.6 | 18.7 ±34.7 | 6.1 ±40.6 | -384.1 ±115.2 | -416 ±123.4 |
| SMLDG (Li et al., 2020) | -2765.9 ±743.8 | -2591.8 ±1013.5 | -12.7 ±39.9 | -134.8 ±97.8 | -601.8 ±283.9 | -623.0 ±329 |
| MLDG (Li et al., 2018) | -10509.2 ±662.2 | -9508.4 ±580.8 | -76.2 ±122.3 | -162.4 ±201.6 | -1164.1 ±511.0 | -1386.9 ±724.8 |
| QMIX-Attention (Iqbal et al., 2021) | -1252.3 ±75.0 | -1202.8 ±137.9 | 34.7 ±20.1 | 2.1 ±17.0 | -305.3 ±9.7 | -347.9 ±15.7 |
| QMIX-MLP (Rashid et al., 2018) | -1464.6 ±136.2 | -1325.1 ±177.9 | 17.3 ±34.3 | -8.1 ±13.2 | -337.8 ±52.7 | -357.5 ±14.4 |

Green: ours ; Yellow: MARL methods ; Red: other model-agnostic methods ; Blue: MARL backbones

**Bold** represents the best result across column, and underlined represents the second-best

penalized for colliding with the targets. (5) **Repel** is a two-entity (agents, adversaries) environment where agents try to maximize their distance from adversaries without leaving the play region. The play region is defined for each environment in Appendix B. (6) **Hunt** is a two-entity (agents, adversaries) environment where agents hunt down the adversaries. The adversaries (deterministically) move away from the agents at each time step, therefore requiring agents to cooperate intelligently.

## 5 EMPIRICAL STUDY

**Baselines and Setting.** We examine the empirical performance of FLICKERFUSION against 11 relevant baselines. The baselines are divided into 3 categories: **(i)** MARL backbone methods, **(ii)** MARL backbone methods with model-agnostic domain generalization methods, and **(iii)** recent relevant MARL methods. **(i)** The two backbone architecture we use are QMIX-MLP (Rashid et al., 2018) and QMIX-Attention (Iqbal et al., 2021). **(ii)** To ensure that our method is most competitive vis-à-vis domain generalization, we implement four model-agnostic domain generalization methods on top of the best performing backbone, QMIX-Attention. These are MLDG (Li et al., 2018), SMLDG (Li et al., 2020), DG-MAML (Wang et al., 2021a), and Meta DotProd (Kedia et al., 2021). More recent methods are highly domain- and model-specific, leading to logically flawed implementations if applied to our problem setting. We are the *first* to enact such rigor in empirically examining model-agnostic domain generalization methods in the MARL problem setting. **(iii)** The remaining five baselines are most relevant to our problem setting. UPDeT (Hu et al., 2021), REFIL (Iqbal et al., 2021), CAMA (Shao et al., 2023), and ODIS (Zhang et al., 2023a) are representative methods that focus on dynamic team composition. CAMA and ODIS *specifically* study zero-shot OOD generalization and show state-of-the-art results. Finally, we include the most recently proposed state-of-the-art general MARL method, ACORM (Hu et al., 2024).

While related, we do not include Wang et al. (2023a) and Tian et al. (2023). Wang et al. (2023a)'s implementation is *complex*, *closed-source*, and only empirically examined with a PPO (Schulman et al., 2017) backbone. On-policy RL backbones are notoriously expensive to train for MARL, resulting in poor scalability (Papoudakis et al., 2021). Similarly, Tian et al. (2023)'s implementation is *complex*, *closed-source*, with only *two* baselines and no study against their backbone architecture.

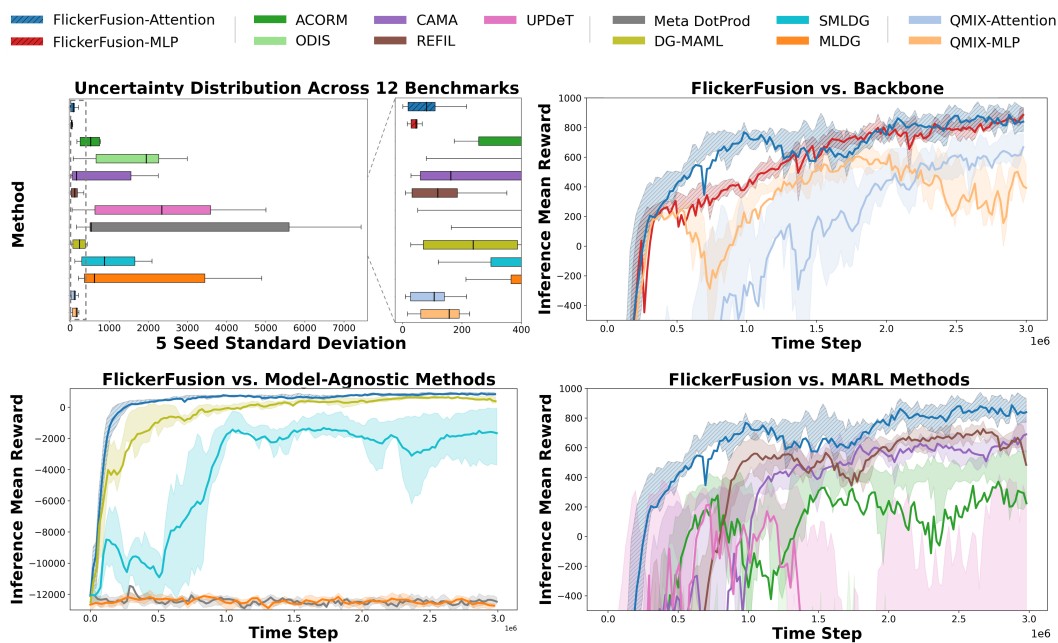

Figure 5: Additional empirical study visualizations. Top-left is the box-and-whisker plot uncertainty distributions across methods. The remainder are reward curves of Repel (OOD1), demonstrating that FLICKERFUSION (**dark blue**, **red**) not only performs well at 3M steps, but also converges faster. The shaded regions indicate quartile 1 to 3—complementing the top-left standard deviation statistics.

For readers that are interested in improved generalization through multi-task learning, they can refer to Qin et al. (2024).

We average results over 5 seeds per ⟨benchmark, method⟩ pair. These seeds are *randomly* picked and *never* changed. For fair comparison, the *same* degree-of-freedom and sample size are used for hyperparameter tuning on each ⟨benchmark, method⟩—reported in Appendix C. Overlapping hyperparameters that may significantly influence performance are equalized across methods. Each seed is trained for 3 million steps, and the 8-sample mean inference reward curve is recorded. Training steps were determined *a priori* and never changed. Following Agarwal et al. (2021); Bettini et al. (2024), we also report the inter-quartile mean (IQM) and the 95% inter-quartile range (IQR) uncertainty distribution in Appendix D, which do not diverge from main text findings.

**Results.** Table 1 presents the overall results of final mean rewards. FLICKERFUSION ranks first in 10 out of 12 benchmarks, and it always ranks at least second except for Adversary (OOD2). Using our FLICKERFUSION method with QMIX-MLP and QMIX-Attention leads to improvements 22 out of 24 times compared to not using it, highlighting the effectiveness of FLICKERFUSION as a promising plug-in method for OOD MARL. Fig. 5 (top-left) illustrates the box-and-whisker plot of the uncertainty of each method across all benchmarks. Observe how FLICKERFUSION achieves the lowest median uncertainty relative to baselines, suggesting that it is the most robust OOD MARL method across different environments. For instance, Meta DotProd (grey), despite achieving good performance in Adversary, displays high uncertainty across the considered environments. This renders its one-off superior performance in Adversary pragmatically meaningless. Another observation is that FLICKERFUSION improves agent-wise policy heterogeneity, further detailed in Appendix F.

We also present the full reward curves over the entire horizon for Repel (OOD1) in Fig. 5. Note that ODIS's reward curve is unavailable for the first million steps as it includes a pre-training stage (Berg et al., 2023), and even after that stage, it is still not visible due to its poor performance. The remaining 11 benchmarks' results are presented in Appendix E.

**Ablation – DAED vs. ED.** We investigate the effect of being domain-aware when doing ED. FLICKERFUSION without **DA** (domain-aware) is implemented by replacing line 6 of Alg. 2 with

Table 2: Ablation empirical results, 3M steps (final inference reward)

| MPEv2 Environment | Spread | | Repel | | Tag | | Guard | | Adversary | | Hunt | |
|---|---|---|---|---|---|---|---|---|---|---|---|---|
| Method | OOD1 | OOD2 | OOD1 | OOD2 | OOD1 | OOD2 | OOD1 | OOD2 | OOD1 | OOD2 | OOD1 | OOD2 |
| FLICKERFUSION-Attention | **-198.4** | **-230.1** | 845.6 | 1,070.2 | **-2.1** | **-9.4** | **-1,258.3** | -1,160.0 | **60.9** | **9.9** | **-297.1** | -337.5 |
| wo. DA | -212.5 | -240.2 | **854.1** | **1,114.9** | -6.0 | -21.2 | -1,410.3 | **-1,158.4** | 46.1 | -17.1 | -316.0 | **-333.0** |
| FLICKERFUSION-MLP | **-183.7** | **-209.0** | 906.1 | 1,189.0 | -18.8 | -33.7 | **-1,127.2** | **-962.9** | 56.3 | 9.3 | -278.5 | -305.3 |
| wo. DA | -196.0 | -222.4 | 577.8 | 1,046.0 | -204.9 | -427.9 | -1,447.6 | -1,184.5 | 43.7 | 3.3 | -285.6 | -325.1 |

**Bold** represents the best result across backbone type and column

$\Delta_t^{inf} \leftarrow$ Uniform$(\mathbf{0}, \mathbf{N}^{inf})$, i.e., the number of dropped entities is determined *independently* from the in-domain entity composition $\mathbf{N}^{train}$. As seen in Table 2, **DA** dramatically improves performance under the MLP backbone, but not as much under the attention backbone. This is consistent with the discussion in Sec. 3.1. Since QMIX-Attention has a greater level of inductive bias encoded into $\boldsymbol{\theta}_Q'$ via $\phi^e$ than QMIX-MLP, the effect from **DA** is relatively weaker.

**Compute.** Finally, we examine the training compute overhead caused by FLICKERFUSION. When training 3 million steps, on an RTX 3090 24GB and 15 core machine, QMIX-MLP→FLICKERFUSION-MLP and QMIX-Attention→FLICKERFUSION-Attention only incurs an additional $4646.4s \rightarrow 4838.3s$ ($+4.1\%$) and $5560.5s \rightarrow 6066.1s$ ($+9.1\%$) run-time cost, respectively. Moreover, memory cost is *reduced* as additional $\boldsymbol{\theta}_Q'$ is not needed.

## 6 DISCUSSION AND TAKEAWAYS

**Novel universal inductive bias injection.** We show a new inductive bias encoding method for OOD MARL. We rigorously investigate the cause of poor performance under OOD. Then, we systematically identify an orthogonal approach. Our findings highlight that OOD MARL problems are better solved via input augmentation, rather than encoding priors into new parameters. Our approach is also *universal* in that it can easily apply to both MLP and attention backbones.

**Potential in non-attention backbones.** Unlike other MARL generalization methods that *only* work with an attention backbone (Shao et al., 2023; Zhang et al., 2023a), FLICKERFUSION works exceptionally well under MLP *and* attention architectures. Surprisingly, FLICKERFUSION-MLP often beats all other existing attention-based methods, even with smaller model (parameter) size (Table 1). Further, during hyperparameter tuning, *all else equal*, we find no meaningful relationship between model size (parameters) and performance. This echoes other domain literature, which shows that attention may not always be the most optimal design choice (Liu et al., 2021; Zeng et al., 2023).

**No reward-uncertainty trade-off.** A salient advantage of FLICKERFUSION is its significant reduction in uncertainty in OOD inference relative to baselines (Fig. 5 (top-left)). Remarkably, FLICKERFUSION is the *only* method where the uncertainty decreases relative to backbone. Also, interestingly, we observe a negative relationship between reward performance and uncertainty. Meaning, better methods vis-à-vis reward ($\uparrow$) is also on average better in terms of uncertainty ($\downarrow$). We provide scatter plots and linear curve fitting analysis in Appendix G.

**Online over offline data.** While ODIS uses an additional one million offline data set generating step which results in significant computational overhead, it performs poorly. While it is important to note that ODIS may be useful when abundant offline data is available *a priori*, it does not work well under our problem setting. In short, self-supervised pre-training from offline data for MARL domain generalization should be used cautiously, especially if offline data is not already available.

**Naïve model-agnostic approaches are ill-advised for MARL.** Unfortunately, we do not observe any meaningful improvement in performance when implementing model-agnostic domain generalizing methods (see the red part in Table 1). On the contrary, the worst performers in terms of reward and uncertainty come from this pool of methods. However, this is not surprising as these methods were not created with MARL tasks in mind.

ACKNOWLEDGMENTS

This work was supported by Institute of Information & Communications Technology Planning & Evaluation (IITP) grants funded by the Korean government (MSIT) (No.2019-0-00075, Artificial Intelligence Graduate School Program (KAIST); No. 2022-0-00871, Development of AI Autonomy and Knowledge Enhancement for AI Agent Collaboration).

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

CONTENTS

## A    PROOF OF PROPOSITION 3.1

(We follow the proof for sample complexity of learning discrete distribution as in a note by C. Canonne)

Note that $d(i) = \sum_{a \in [N_A]} \mathbb{1}[i \in D_a]$, where $D_a \in \binom{[N_\ell]}{\Delta_\ell}$ is the random entity subset of size $\Delta_\ell$ to be dropped by agent $a$. For each $i \in [N_\ell]$, $\mathbb{1}[i \in D_a] \sim \text{Ber}(\Delta_\ell/N_\ell)$, and thus, $d(i) \sim \text{Bin}(N_A, \Delta_\ell/N_\ell)$. We then conclude as follows:

$$\mathbb{E}\left[\sum_{i \in [N_\ell]} \left|\frac{d(i)}{N_A} - \frac{\Delta_\ell}{N_\ell}\right|\right] \leq \sum_{i \in [N_\ell]} \sqrt{\mathbb{E}\left[\left(\frac{d(i)}{N_A} - \frac{\Delta_\ell}{N_\ell}\right)^2\right]} \qquad \text{(Jensen's inequality)}$$

$$= \frac{1}{N_A} \sum_{i \in [N_\ell]} \sqrt{\mathbb{E}\left[\left(d(i) - \frac{\Delta_\ell N_A}{N_\ell}\right)^2\right]}$$

$$= \frac{1}{N_A} \sum_{i \in [N_\ell]} \sqrt{N_A \frac{\Delta_\ell}{N_\ell}\left(1 - \frac{\Delta_\ell}{N_\ell}\right)} \qquad (d(i) \sim \text{Bin}(N_A, \Delta_\ell/N_\ell))$$

$$= \sqrt{\frac{\Delta_\ell(N_\ell - \Delta_\ell)}{N_A}}.$$

$\square$

## B  DEFERRED DETAILS FOR MPEv2

### B.1  COMMON FEATURES OF MPEv2

In MPEv2, entities are placed in a 2-dimensional continuous environment. Distances within this environment are computed using Euclidean metrics. The distance $d(e_i, e_j)$ between two entities $e_i$ and $e_j$ with radii $r_i \in \mathbb{R}$ and $r_j \in \mathbb{R}$ is computed as:

$$d(e_i, e_j) = \|\mathbf{p}_i - \mathbf{p}_j\|_2 - r_i - r_j \tag{A1}$$

where $\mathbf{p}_i \in \mathbb{R}^2$ and $\mathbf{p}_j \in \mathbb{R}^2$ are position vectors. The play areas are square and centered at the origin $[0, 0]$. The coordinates are constrained within the range $x \in [-B, B]$ and $y \in [-B, B]$, where $B$ is the half-width of the play area. Each agent can select from five discrete actions at every timestep. Agents receive a global reward and observe the global state.

| Action Shape | $u \in \mathcal{U} \in \mathbb{N}^5$ |
|---|---|
| Action Space | `[no_action, move_left, move_right, move_down, move_up]` |

Table 3: Action space for agents in MPEv2

The notation `vector(n)` denotes a vector of size n. $\oplus$ denotes vector concatenation. $U(\cdot, \cdot)$ denotes uniform sampling. $\delta_{i,j}$ is the Kronecker delta function, defined as $\delta_{i,j} = \begin{cases} 1 & \text{if } i = j \\ 0 & \text{if } i \neq j \end{cases}$.

### B.2  TAG

This environment features $n_a$ agents and $n_{adv}$ heuristic adversaries that move towards the closest agent. The primary objective of this scenario is for the agents to learn to avoid collisions with adversaries while remaining within the defined boundaries.

| Entity Types | `[agent, adversary]` |
|---|---|
| Observation Space | `[(entity_type(2) ⊕ entity_pos(2) ⊕ entity_vel(2))*(`$n_a$` + `$n_{adv}$`) ⊕ last_action(5)]` |
| Episode Timesteps ($t_{max}$) | 200 |
| Play Area | $\{\langle x, y \rangle : x \in [-2, 2], y \in [-2, 2]; x, y \in \mathbb{R}\}$ |

Table 4: Tag environment properties

Let $\mathcal{E}_a$ be the set of agents and $\mathcal{E}_{adv}$ be the set of adversaries. The reward for each $t$ for this environment is given as:

$$R(\cdot) := \sum_{a \in \mathcal{E}_a} \left( \mathbb{1}_{outside}(x_a, y_a) \cdot \max\left(-25, \min\left(-5^{|x_a|-2}, -5^{|y_a|-2}\right)\right) - 2 \cdot \sum_{adv \in \mathcal{E}_{adv}} \mathbb{1}_{collision}(a, adv) \right). \tag{A2}$$

Agents receive a penalty of -2 for each collision with an adversary and an additional penalty for traveling beyond the boundaries, with the total boundary penalty capped at -25.

| | $n_a^{init}$ | $n_{adv}^{init}$ | $n_a^{intra}$ | $n_{adv}^{intra}$ |
|---|---|---|---|---|
| **In-Domain** | $U(1,3)$ | $U(1,3)$ | 1 | 1 |
| **OOD1** | 5 | 5 | $U(1,2)$ | 0 |
| **OOD2** | 5 | 5 | 0 | $U(1,2)$ |

Table 5: Number of entities for each domain

$n_a^{init}$ and $n_{adv}^{init}$ represents the initial quantities for agents and adversaries, respectively. $n_a^{intra}$ and $n_{adv}^{intra}$ denotes the number of entities added intra-trajectory. These variables are defined either as fixed values or as ranges from which the number of entities can be randomly sampled.

| | **Agent** | **Adversary** |
|---|---|---|
| $\boldsymbol{x_{init}}$ | $U(-0.25B, 0.25B)$ | $U(-B, -0.6B)$ |
| $\boldsymbol{y_{init}}$ | $U(-0.25B, 0.25B)$ | $U(0.6B, B)$ |
| $\boldsymbol{x_{intra}}$ | $B \cdot [U(-1,1) \cdot (\delta_{\epsilon,0} + \delta_{\epsilon,2}) + \delta_{\epsilon,1} - \delta_{\epsilon,3}]$ | $B \cdot [U(-1,1) \cdot (\delta_{\epsilon,0} + \delta_{\epsilon,2}) + \delta_{\epsilon,1} - \delta_{\epsilon,3}]$ |
| $\boldsymbol{y_{intra}}$ | $B \cdot [U(-1,1) \cdot (\delta_{\epsilon,1} + \delta_{\epsilon,3}) + \delta_{\epsilon,0} - \delta_{\epsilon,2}]$ | $B \cdot [U(-1,1) \cdot (\delta_{\epsilon,1} + \delta_{\epsilon,3}) + \delta_{\epsilon,0} - \delta_{\epsilon,2}]$ |

Table 6: Entity spawning location

The position of entity at the beginning is given as $\mathbf{p}_e := [x_{init}, y_{init}]$ and the position of entity introduced during the trajectory is given as $\mathbf{p}_e := [x_{intra}, y_{intra}]$. Agents are initially placed within a smaller square centered around the origin and adversaries are distributed along the corner of the second quadrant using uniform random sampling. The intra-trajectory spawning location $\mathbf{p}_e$ for agents and adversaries is determined by randomly selecting one of the four edges $\epsilon \in \{0, 1, 2, 3\}$ of a square boundary.

### B.3 SPREAD

This environment consists of $n_a$ agents and $n_{tar}$ target landmarks. The agents are trained to effectively distribute themselves among targets.

| **Entity Types** | `[agent, target]` |
|---|---|
| **Observation Space** | `[(entity_type(2) ⊕ entity_pos(2) ⊕ entity_vel(2))*(n_a + n_tar) ⊕ last_action(5)]` |
| **Episode Timesteps** $(t_{max})$ | 100 |
| **Play Area** | $\{\langle x, y \rangle : x \in [-1,1], y \in [-1,1]; x,y \in \mathbb{R}\}$ |

Table 7: Spread Environment Properties

Let $\mathcal{E}_a$ be the set of agents and $\mathcal{E}_{tar}$ be the set of target landmarks. Agents are rewarded for each $t$ based on the proximity of each target to its nearest agent such that:

$$R(\cdot) := - \sum_{tar \in \mathcal{E}_{tar}} \min_{a \in \mathcal{E}_a} (d(a, tar)). \tag{A3}$$

|  | $n_a^{init}$ | $n_{tar}^{init}$ | $n_a^{intra}$ | $n_{tar}^{intra}$ |
|---|---|---|---|---|
| **In-Domain** | $U(1,3)$ | $U(1,3)$ | 1 | 1 |
| **OOD1** | 5 | 5 | $U(1,2)$ | 0 |
| **OOD2** | 5 | 5 | 0 | $U(1,2)$ |

Table 8: Number of Entities for Each Domain

$n_a^{init}$ and $n_{tar}^{init}$ represent the initial quantities for agents and targets, respectively. $n_a^{intra}$ and $n_{tar}^{intra}$ denote the number of entities added intra-trajectory. These variables are defined either as fixed values or as ranges from which the number of entities can be randomly sampled.

|  | **Agent** | **Target** |
|---|---|---|
| $x_{init}$ | $U(-B,B)$ | $U(-B,B)$ |
| $y_{init}$ | $U(-B,B)$ | $U(-B,B)$ |
| $x_{intra}$ | $B \cdot [U(-1,1) \cdot$ $(\delta_{\epsilon,0} + \delta_{\epsilon,2}) + \delta_{\epsilon,1} - \delta_{\epsilon,3}]$ | $B \cdot [U(-1,1) \cdot$ $(\delta_{\epsilon,0} + \delta_{\epsilon,2}) + \delta_{\epsilon,1} - \delta_{\epsilon,3}]$ |
| $y_{intra}$ | $B \cdot [U(-1,1) \cdot$ $(\delta_{\epsilon,1} + \delta_{\epsilon,3}) + \delta_{\epsilon,0} - \delta_{\epsilon,2}]$ | $B \cdot [U(-1,1) \cdot$ $(\delta_{\epsilon,1} + \delta_{\epsilon,3}) + \delta_{\epsilon,0} - \delta_{\epsilon,2}]$ |

Table 9: Entity spawning location

The position of entity at the beginning is given as $\mathbf{p}_e = [x_{init}, y_{init}]$ and the position of entity introduced during the trajectory is given as $\mathbf{p}_e = [x_{intra}, y_{intra}]$. The intra-trajectory spawning location $\mathbf{p}_e$ for agents and landmarks is determined by randomly selecting one of the four edges $\epsilon \in \{0, 1, 2, 3\}$ of a square boundary.

## B.4 GUARD

This environment has $n_a$ agents and $n_{tar}$ heuristic targets. The objective is to minimize the distance between each target and its closest agents. Targets move in random directions every 50 timesteps, while remaining within the boundaries. At a high level, the agents must learn to organize into groups to cover and track the moving targets.

| **Entity Types** | `[agent, target]` |
|---|---|
| **Observation Space** | `[(entity_type(2) ⊕ entity_pos(2) ⊕ entity_vel(2))*(n_a + n_tar) ⊕ last_action(5)]` |
| **Episode Timesteps** ($t_{max}$) | 200 |
| **Play Area** | $\{\langle x,y \rangle : x \in [-2,2], y \in [-2,2]; x, y \in \mathbb{R}\}$ |

Table 10: Guard Environment Properties

Let $\mathcal{E}_a$ be the set of agents and $\mathcal{E}_{tar}$ be the set of heuristic targets. Then, the reward for each $t$ for this environment is given as:

$$R(\cdot) := - \sum_{tar \in \mathcal{E}_{tar}} \sum_{k=1}^{i} \left( \min_{a \in \mathcal{E}_a} d(a, tar) \right)_k - 5 \cdot \mathbb{1}_{collision}(a, tar) \quad \text{where} \quad i = \left\lceil \frac{n_a}{n_{tar}} \right\rceil. \quad (A4)$$

Agents are rewarded based on the distances of each target to its $i$-closest agents, and are penalized for collisions with targets.

| | $n_a^{init}$ | $n_{tar}^{init}$ | $n_a^{intra}$ | $n_{tar}^{intra}$ |
|---|---|---|---|---|
| **In-Domain** | $U(1,3)$ | $U(1,2)$ | 1 | 1 |
| **OOD1** | 5 | 4 | $U(1,3)$ | 0 |
| **OOD2** | 5 | 4 | 0 | $U(1,2)$ |

Table 11: Number of Entities for Each Domain

$n_a^{init}$ and $n_{tar}^{init}$ represent the initial quantities for agents and targets, respectively. For in-domain training and testing, these quantities can vary within a specified range, which is used to randomly determine the number of agents and targets. $n_a^{intra}$ and $n_{tar}^{intra}$ denote the number of entities added intra-trajectory.

| | **Agent** | **Target** |
|---|---|---|
| $x_{init}$ | $U(-B,B)$ | $U(-0.5B, 0.5B)$ |
| $y_{init}$ | $U(-B,B)$ | $U(-0.5B, 0.5B)$ |
| $x_{intra}$ | $B \cdot [U(-1,1) \cdot (\delta_{\epsilon,0} + \delta_{\epsilon,2}) + \delta_{\epsilon,1} - \delta_{\epsilon,3}]$ | $B \cdot [U(-1,1) \cdot (\delta_{\epsilon,0} + \delta_{\epsilon,2}) + \delta_{\epsilon,1} - \delta_{\epsilon,3}]$ |
| $y_{intra}$ | $B \cdot [U(-1,1) \cdot (\delta_{\epsilon,1} + \delta_{\epsilon,3}) + \delta_{\epsilon,0} - \delta_{\epsilon,2}]$ | $B \cdot [U(-1,1) \cdot (\delta_{\epsilon,1} + \delta_{\epsilon,3}) + \delta_{\epsilon,0} - \delta_{\epsilon,2}]$ |

Table 12: Entity spawning location

The position of entity at the beginning is given as $\mathbf{p}_e = [x_{init}, y_{init}]$ and the position of entity introduced during the trajectory is given as $\mathbf{p}_e = [x_{intra}, y_{intra}]$. Agents and targets are initially distributed across the play area using uniform random sampling. The intra-trajectory spawning location $\mathbf{p}_e$ for agents and targets is determined by randomly selecting one of the four edges $\epsilon \in \{0, 1, 2, 3\}$ of a square boundary.

## B.5 REPEL

This environment consists of $n_a$ agents and $n_{adv}$ heuristic adversaries which move towards the closest agent. Agents are trained to maximize their distance from adversaries while staying within defined boundaries.

| **Entity Types** | `[agent, adversary]` |
|---|---|
| **Observation Space** | `[(entity_type(2) ⊕ entity_pos(2) ⊕ entity_vel(2))*(`$n_a$` + `$n_{adv}$`) ⊕ last_action(5)]` |
| **Episode Timesteps** ($t_{max}$) | 150 |
| **Play Area** | $\{\langle x,y \rangle : x \in [-2.5, 2.5], y \in [-2.5, 2.5]; x,y \in \mathbb{R}\}$ |

Table 13: Repel Environment Properties

Let $\mathcal{E}_a$ be the set of agents and $\mathcal{E}_{adv}$ be the set of adversaries. The reward for each $t$ for this environment is given as:

$$R(\cdot) := \sum_{a \in \mathcal{E}_a} \left[ \mathbb{1}_{outside}(x_a, y_a) \cdot \max\left( -25, \min\left( -5^{|x_a|-2}, -5^{|y_a|-2} \right) \right) \right] + \sum_{adv \in \mathcal{E}_{adv}} \min_{a \in \mathcal{E}_a} d(a, adv).$$

$$(A5)$$

Thus, agents are rewarded based on how far the nearest agent is to each adversary and receive a penalty for traveling beyond the boundaries, with the total boundary penalty capped at -25.

| | $n_a^{init}$ | $n_{adv}^{init}$ | $n_a^{intra}$ | $n_{adv}^{intra}$ |
|---|---|---|---|---|
| **In-Domain** | $U(1,3)$ | $U(1,3)$ | 1 | 1 |
| **OOD1** | 5 | 5 | $U(1,2)$ | 0 |
| **OOD2** | 5 | 5 | 0 | $U(1,2)$ |

Table 14: Number of Entities for Each Domain

$n_a^{init}$ and $n_{adv}^{init}$ represent the initial quantities for agents and adversaries, respectively. For in-domain training and testing, these quantities can vary within a specified range, which is used to randomly determine the number of agents and adversaries. $n_a^{intra}$ and $n_{adv}^{intra}$ denote the number of entities added intra-trajectory.

| | **Agent** | **Adversary** |
|---|---|---|
| $x_{init}$ | $U(-B,B)$ | $U(-0.2,0.2)$ |
| $y_{init}$ | $U(-B,B)$ | $U(-0.2,0.2)$ |
| $x_{intra}$ | $B \cdot [U(-1,1) \cdot$ $(\delta_{\epsilon,0} + \delta_{\epsilon,2}) + \delta_{\epsilon,1} - \delta_{\epsilon,3}]$ | $B \cdot [U(-1,1) \cdot$ $(\delta_{\epsilon,0} + \delta_{\epsilon,2}) + \delta_{\epsilon,1} - \delta_{\epsilon,3}]$ |
| $y_{intra}$ | $B \cdot [U(-1,1) \cdot$ $(\delta_{\epsilon,1} + \delta_{\epsilon,3}) + \delta_{\epsilon,0} - \delta_{\epsilon,2}]$ | $B \cdot [U(-1,1) \cdot$ $(\delta_{\epsilon,1} + \delta_{\epsilon,3}) + \delta_{\epsilon,0} - \delta_{\epsilon,2}]$ |

Table 15: Entity spawning location

The position of entity at the beginning is given as $\mathbf{p}_e = [x_{init}, y_{init}]$ and the position of entity introduced during the trajectory is given as $\mathbf{p}_e = [x_{intra}, y_{intra}]$. Adversaries are initially placed within a smaller square centered around the origin and agents are distributed across the play area using uniform random sampling. The intra-trajectory spawning location $\mathbf{p}_e$ for agents and adversaries is determined by randomly selecting one of the four edges $\epsilon \in \{0, 1, 2, 3\}$ of a square boundary.

## B.6 ADVERSARY

This environment features $n_a$ agents, $n_{adv}$ heuristic adversaries, $n_{tar}$ target landmarks, and $n_{dec}$ decoy landmarks. Adversaries cannot distinguish between target and decoy landmarks, and move towards the landmark closest to any agent. At a high level, the agents must learn to stay close to the targets while guiding adversaries away from them.

| **Entity Types** | `[agent, adversary, target, decoy]` |
|---|---|
| **Observation Space** | `[(entity_type(4) ⊕ entity_pos(2)` `⊕ entity_vel(2))*(`$n_a$` + `$n_{adv}$`) ⊕` `last_action(5)]` |
| **Episode Timesteps** ($t_{max}$) | 150 |
| **Play Area** | $\{\langle x,y \rangle : x \in [-1.3, 1.3], y \in [-1.3, 1.3]; x, y \in \mathbb{R}\}$ |

Table 16: Adversary Environment Properties

Let $\mathcal{E}_a$ be the set of agents, $\mathcal{E}_{adv}$ the set of adversaries, and $\mathcal{E}_{tar}$ the set of target landmarks. The reward for each $t$ for this environment is given as:

$$R(\cdot) := - \sum_{tar \in \mathcal{E}_{tar}} \min_{a \in \mathcal{E}_a} d(a, tar) + \sum_{tar \in \mathcal{E}_{tar}} \min_{adv \in \mathcal{E}_{adv}} d(adv, tar). \qquad \text{(A6)}$$

Hence, agents are rewarded based the proximity of the nearest agent and adversary to each target, or more specifically, the difference between the sum of minimum distances from adversaries to each target and the sum of minimum distances from agents to each target.

| | $n_a^{init}$ | $n_{adv}^{init}$ | $n_{tar}^{init}$ | $n_{dec}^{init}$ | $n_a^{intra}$ | $n_{adv}^{intra}$ | $n_{tar}^{intra}$ | $n_{dec}^{intra}$ |
|---|---|---|---|---|---|---|---|---|
| **In-Domain** | $U(1,3)$ | $U(1,3)$ | $U(1,2)$ | $U(1,2)$ | 1 | 1 | 1 | 1 |
| **OOD1** | 5 | 5 | $U(1,2)$ | $U(1,2)$ | $U(1,2)$ | 0 | 1 | 1 |
| **OOD2** | 5 | 5 | $U(1,2)$ | $U(1,2)$ | 0 | $U(1,2)$ | 1 | 1 |

Table 17: Number of Entities for Each Domain

$n_a^{init}$, $n_{adv}^{init}$, $n_{tar}^{init}$, and $n_{dec}^{init}$ represent the initial quantities for each entity type. This can be given a set value or a range of values used to randomly determine the number of agents and adversaries. $n_a^{intra}$, $n_{adv}^{intra}$, $n_{tar}^{intra}$, and $n_{dec}^{intra}$ denote the number of entities added intra-trajectory.

| | **Agent** | **Adversary** | **Target** | **Decoy** |
|---|---|---|---|---|
| $x_{init}$ | $U(-B, B)$ | $U(-B, B)$ | $U(-B, B)$ | $U(-B, B)$ |
| $y_{init}$ | $U(-B, B)$ | $U(-B, B)$ | $U(-B, B)$ | $U(-B, B)$ |
| $x_{intra}$ | $B \cdot [U(-1,1) \cdot (\delta_{\epsilon,0} + \delta_{\epsilon,2}) + \delta_{\epsilon,1} - \delta_{\epsilon,3}]$ | $B \cdot [U(-1,1) \cdot (\delta_{\epsilon,0} + \delta_{\epsilon,2}) + \delta_{\epsilon,1} - \delta_{\epsilon,3}]$ | $U(-B, B)$ | $U(-B, B)$ |
| $y_{intra}$ | $B \cdot [U(-1,1) \cdot (\delta_{\epsilon,1} + \delta_{\epsilon,3}) + \delta_{\epsilon,0} - \delta_{\epsilon,2}]$ | $B \cdot [U(-1,1) \cdot (\delta_{\epsilon,1} + \delta_{\epsilon,3}) + \delta_{\epsilon,0} - \delta_{\epsilon,2}]$ | $U(-B, B)$ | $U(-B, B)$ |

Table 18: Entity spawning location

The position of entity at the beginning is given as $\mathbf{p}_e = [x_{init}, y_{init}]$ and the position of entity introduced during the trajectory is given as $\mathbf{p}_e = [x_{intra}, y_{intra}]$. All entities are initially distributed across the play area using uniform random sampling. The intra-trajectory spawning location $\mathbf{p}_e$ for agents and adversaries is determined by randomly selecting one of the four edges $\epsilon \in \{0, 1, 2, 3\}$ of a square boundary.

## B.7 HUNT

This environment consists of $n_a$ agents and $n_{adv}$ heuristic adversaries which move away from the closest agent while staying within defined boundaries. Agents are trained to minimize their distance from adversaries.

| Entity Types | [agent, adversary] |
|---|---|
| **Observation Space** | [(entity_type(2) ⊕ entity_pos(2) ⊕ entity_vel(2))*($n_a$ + $n_{adv}$) ⊕ last_action(5)] |
| **Episode Timesteps ($t_{max}$)** | 200 |
| **Play Area** | $\{\langle x, y \rangle : x \in [-2, 2], y \in [-2, 2]; x, y \in \mathbb{R}\}$ |

Table 19: Hunt Environment Properties

Let $\mathcal{E}_a$ be the set of agents and $\mathcal{E}_{adv}$ be the set of adversaries. The reward for each $t$ for this environment is given as:

$$R(\cdot) := -\left( \frac{1}{n_{adv}} \sum_{adv \in \mathcal{E}_{adv}} \min_{a \in \mathcal{E}_a} d(a, adv) + \frac{1}{n_a} \sum_{a \in \mathcal{E}_a} \min_{adv \in \mathcal{E}_{adv}} d(a, adv) \right). \qquad (A7)$$

Thus, agents are rewarded based on are rewarded based on the average minimum distances to the nearest adversaries, as well as the average minimum distances from adversaries to the nearest agents.

| | $n_a^{init}$ | $n_{adv}^{init}$ | $n_a^{intra}$ | $n_{adv}^{intra}$ |
|---|---|---|---|---|
| **In-Domain** | $U(1,3)$ | $U(1,3)$ | 1 | 1 |
| **OOD1** | 5 | 5 | $U(1,2)$ | 0 |
| **OOD2** | 5 | 5 | 0 | $U(1,2)$ |

Table 20: Number of Entities for Each Domain

$n_a^{init}$ and $n_{adv}^{init}$ represent the initial quantities for agents and adversaries, respectively. For in-domain training and testing, these quantities can vary within a specified range, which is used to randomly determine the number of agents and adversaries. $n_a^{intra}$ and $n_{adv}^{intra}$ denote the number of entities added intra-trajectory.

| | **Agent** | **Adversary** |
|---|---|---|
| $x_{init}$ | $U(-B, -0.25B) \cup U(0.25B, B)$ | $U(-0.25B, 0.25B)$ |
| $y_{init}$ | $U(-B, -0.25B) \cup U(0.25B, B)$ | $U(-0.25B, 0.25B)$ |
| $x_{intra}$ | $B \cdot [U(-1,1) \cdot (\delta_{\epsilon,0} + \delta_{\epsilon,2}) + \delta_{\epsilon,1} - \delta_{\epsilon,3}]$ | $B \cdot [U(-1,1) \cdot (\delta_{\epsilon,0} + \delta_{\epsilon,2}) + \delta_{\epsilon,1} - \delta_{\epsilon,3}]$ |
| $y_{intra}$ | $B \cdot [U(-1,1) \cdot (\delta_{\epsilon,1} + \delta_{\epsilon,3}) + \delta_{\epsilon,0} - \delta_{\epsilon,2}]$ | $B \cdot [U(-1,1) \cdot (\delta_{\epsilon,1} + \delta_{\epsilon,3}) + \delta_{\epsilon,0} - \delta_{\epsilon,2}]$ |

Table 21: Entity spawning location

The position of entity at the beginning is given as $\mathbf{p}_e = [x_{init}, y_{init}]$ and the position of entity introduced during the trajectory is given as $\mathbf{p}_e = [x_{intra}, y_{intra}]$. Adversaries are initially placed within a smaller square centered around the origin and agents are positioned in discrete regions at the corners of the play area. The intra-trajectory spawning location $\mathbf{p}_e$ for agents and adversaries is determined by randomly selecting one of the four edges $\epsilon \in \{0, 1, 2, 3\}$ of a square boundary.

## C HYPERPARAMETERS

### C.1 COMMON HYPERPARAMETERS

In the tables below are the hyperparameters used for MLP backbone models (Table 22) and attention backbone models (Table 23).

| Parameters | Tag | Spread | Guard | Repel | Adversary | Hunt |
|---|---|---|---|---|---|---|
| Test interval | 20000 | 20000 | 20000 | 20000 | 20000 | 20000 |
| Test episodes | 15 | 15 | 15 | 15 | 15 | 15 |
| Epsilon start | 1 | 1 | 1 | 1 | 1 | 1 |
| Epsilon finish | 0.3 | 0.05 | 0.05 | 0.05 | 0.05 | 0.05 |
| Epsilon anneal steps | 4000000 | 500000 | 500000 | 500000 | 1000000 | 500000 |
| Number of parallel envs | 32 | 8 | 8 | 8 | 8 | 8 |
| Batch size | 32 | 32 | 32 | 32 | 32 | 32 |
| Buffer size | 5000 | 5000 | 5000 | 5000 | 5000 | 5000 |
| Max timesteps | 3000000 | 3000000 | 3000000 | 3000000 | 3000000 | 3000000 |
| Mixing embed dimension | 32 | 32 | 32 | 32 | 32 | 32 |
| Hypernet embed dimension | 128 | 128 | 128 | 128 | 128 | 128 |
| Hypernet activation | abs | abs | abs | abs | abs | abs |
| Learning rate | 0.0005 | 0.0003 | 0.0003 | 0.0003 | 0.0005 | 0.0003 |
| RNN hidden dim | 128 | 64 | 64 | 128 | 128 | 128 |
| RNN input dim (ours) | 128 | 32 | 64 | 128 | 32 | 256 |
| RNN hidden dim (ours) | 128 | 64 | 256 | 128 | 64 | 256 |

Table 22: Hyperparameters (MLP backbone)

| Parameters | Tag | Spread | Guard | Repel | Adversary | Hunt |
|---|---|---|---|---|---|---|
| Test interval | 20000 | 20000 | 20000 | 20000 | 20000 | 20000 |
| Test episodes | 15 | 15 | 15 | 15 | 15 | 15 |
| Epsilon start | 1 | 1 | 1 | 1 | 1 | 1 |
| Epsilon finish | 0.3 | 0.05 | 0.05 | 0.05 | 0.05 | 0.05 |
| Epsilon anneal steps | 4000000 | 500000 | 500000 | 500000 | 1000000 | 500000 |
| Number of parallel envs | 32 | 8 | 8 | 8 | 8 | 8 |
| Batch size | 32 | 32 | 32 | 32 | 32 | 32 |
| Buffer size | 5000 | 5000 | 5000 | 5000 | 5000 | 5000 |
| Max timesteps | 3000000 | 3000000 | 3000000 | 3000000 | 3000000 | 3000000 |
| Mixing embed dimension | 32 | 32 | 32 | 32 | 32 | 32 |
| Hypernet embed dimension | 128 | 128 | 128 | 128 | 128 | 128 |
| Attention embed dimension | 128 | 128 | 128 | 128 | 128 | 128 |
| Hypernet activation | abs | softmax | softmax | softmax | softmax | softmax |
| Learning rate | 0.0003 | 0.0003 | 0.0003 | 0.0003 | 0.0005 | 0.0003 |
| RNN hidden dim | 128 | 64 | 64 | 64 | 128 | 128 |
| RNN input dim (ours) | 128 | 64 | 128 | 128 | 256 | 64 |
| RNN hidden dim (ours) | 128 | 64 | 128 | 512 | 256 | 256 |

Table 23: Hyperparameters (Attention backbone)

## C.2 BASELINE MODEL-SPECIFIC HYPERPARAMETERS

In this section, the hyperparameters used for CAMA (Table 24) and UPDeT (Table 25) models are detailed.

| Parameters | Tag | Spread | Guard | Repel | Adversary | Hunt |
|---|---|---|---|---|---|---|
| Cross Entropy weight | 0.0005 | 0.005 | 0.005 | 0.005 | 0.005 | 0.0005 |
| Club weight | 0.1 | 0.1 | 0.5 | 0.1 | 0.1 | 0.1 |
| Default attention concentration rate | 1 | 1 | 1 | 1 | 1 | 1 |
| Default communication message compression rate | 0.5 | 0.5 | 0.5 | 0.5 | 0.5 | 0.5 |
| Gradient clipping | 10 | 10 | 10 | 10 | 10 | 10 |

Table 24: CAMA Hyperparameters

| Parameters | Tag | Spread | Guard | Repel | Adversary | Hunt |
|---|---|---|---|---|---|---|
| Transformer embed dimension | 64 | 64 | 64 | 64 | 128 | 64 |
| Number of heads | 3 | 4 | 4 | 4 | 4 | 3 |
| Number of blocks | 2 | 2 | 2 | 2 | 2 | 2 |

Table 25: UPDeT Hyperparameters

| Parameters | Tag | Spread | Guard | Repel | Adversary | Hunt |
|---|---|---|---|---|---|---|
| Number of heads | 1 | 1 | 1 | 1 | 1 | 1 |
| Number of blocks | 1 | 1 | 1 | 1 | 1 | 1 |
| $\alpha$ | 5.0 | 5.0 | 5.0 | 5.0 | 5.0 | 5.0 |
| $\beta$ | 0.001 | 0.001 | 0.001 | 0.001 | 0.001 | 0.001 |
| $\lambda$ | 5.0 | 5.0 | 5.0 | 5.0 | 5.0 | 5.0 |
| Steps of skill discovery | 1000000 | 1000000 | 1000050 | 1000050 | 1000050 | 1000000 |
| Training steps of QMIX for extracting offline data | 1000000 | 1000000 | 1000000 | 1000000 | 1000000 | 1000000 |
| Skill dimension | 3 | 3 | 3 | 3 | 4 | 3 |

Table 26: ODIS Hyperparameters

# D    ADDITIONAL METRICS

See Table 27 and for IQM values. Furthermore, see Fig. 6 for the 95% IQM uncertainty figure.

Table 27: Empirical results, 3M steps (IQM final inference reward)

| MPEv2 Environment | Spread | | Repel | | Tag | |
|---|---|---|---|---|---|---|
| Method | OOD1 | OOD2 | OOD1 | OOD2 | OOD1 | OOD2 |
| FLICKERFUSION-ATTENTION | -193.0 | -224.1 | 784.2 | _1141.3_ | **-2.92** | **-11.70** |
| FLICKERFUSION-MLP | _-190.8_ | _-223.7_ | **912.0** | **1204.6** | -20.23 | -58.55 |
| ACORM | -229.5 | -323.4 | 559.3 | 418.0 | -241.22 | -260.37 |
| ODIS | -246.7 | -343.1 | -2686.9 | -3291.3 | -1273.53 | -1300.17 |
| CAMA | -198.0 | -283.5 | _787.1_ | 1005.5 | -14.08 | -18.55 |
| REFIL | -207.5 | -300.3 | 549.0 | 877.2 | _-3.11_ | _-12.09_ |
| UPDET | -208.5 | -266.5 | -240.0 | -350.3 | -135.24 | -136.67 |
| Meta DotProd | -407.9 | -642.8 | -12447.5 | -9302.7 | -6186.50 | -5793.31 |
| DG-MAML | -211.0 | -254.9 | 593.6 | 770.8 | -39.12 | -53.81 |
| SMLDG | -248.0 | -303.2 | -161.0 | 88.5 | -182.94 | -130.53 |
| MLDG | -417.1 | -623.0 | -12665.1 | -9750.1 | -193.37 | -162.54 |
| QMIX-ATTENTION | -192.6 | -267.9 | 752.6 | 1008.9 | -14.90 | -25.32 |
| QMIX-MLP | **-189.4** | **-216.9** | 466.8 | 1016.5 | -39.88 | -56.07 |

| MPEv2 Environment | Guard | | Adversary | | Hunt | |
|---|---|---|---|---|---|---|
| Method | OOD1 | OOD2 | OOD1 | OOD2 | OOD1 | OOD2 |
| FLICKERFUSION-ATTENTION | -1280.2 | _-1143.2_ | _63.9_ | 6.9 | _-297.1_ | _-340.3_ |
| FLICKERFUSION-MLP | **-1127.0** | **-967.2** | 53.1 | 8.8 | **-277.1** | **-306.1** |
| ACORM | -2109.8 | -2076.8 | 61.1 | 12.6 | -339.6 | -388.8 |
| ODIS | -1458.3 | -1332.0 | -56.3 | -172.7 | -654.6 | -625.7 |
| CAMA | -4994.2 | -4328.6 | 52.1 | _22.7_ | -1002.7 | -1168.4 |
| REFIL | -1445.5 | -1261.0 | 17.9 | -20.9 | -306.8 | -346.1 |
| UPDET | -1820.7 | -1559.6 | -60.7 | -1.3 | -321.3 | -343.0 |
| Meta DotProd | -7462.2 | -7705.7 | **79.9** | **33.9** | -870.4 | -977.2 |
| DG-MAML | -1550.0 | -1561.5 | 26.4 | 18.0 | -334.0 | -357.3 |
| SMLDG | -2617.3 | -2269.9 | -10.5 | -130.1 | -470.6 | -470.9 |
| MLDG | -10570.8 | -9552.6 | -68.1 | -131.2 | -1208.3 | -1470.7 |
| QMIX-ATTENTION | -1412.6 | -1356.2 | 39.9 | -4.0 | -312.5 | -355.5 |
| QMIX-MLP | _-1251.3_ | -1161.0 | 33.8 | 2.3 | -304.6 | -343.9 |

Green: ours ;  Yellow: MARL methods ;  Red: other model-agnostic methods ;  Blue: MARL backbones
**Bold** represents the best result across column, and underlined represents the second-best

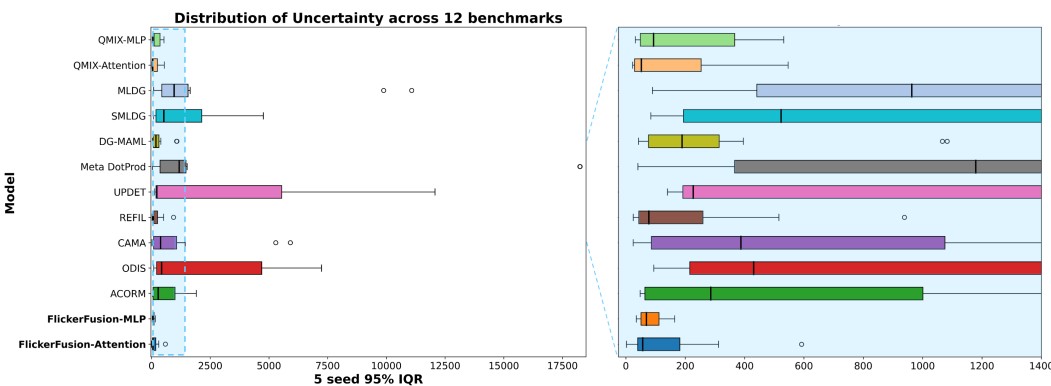

Figure 6: Box and whisker plots aggregated across all benchmarks

# E   EMPIRICAL STUDY (EXTENDED)

The figures below visualize the empirical study results (Figure 7 to 12) for each environment.

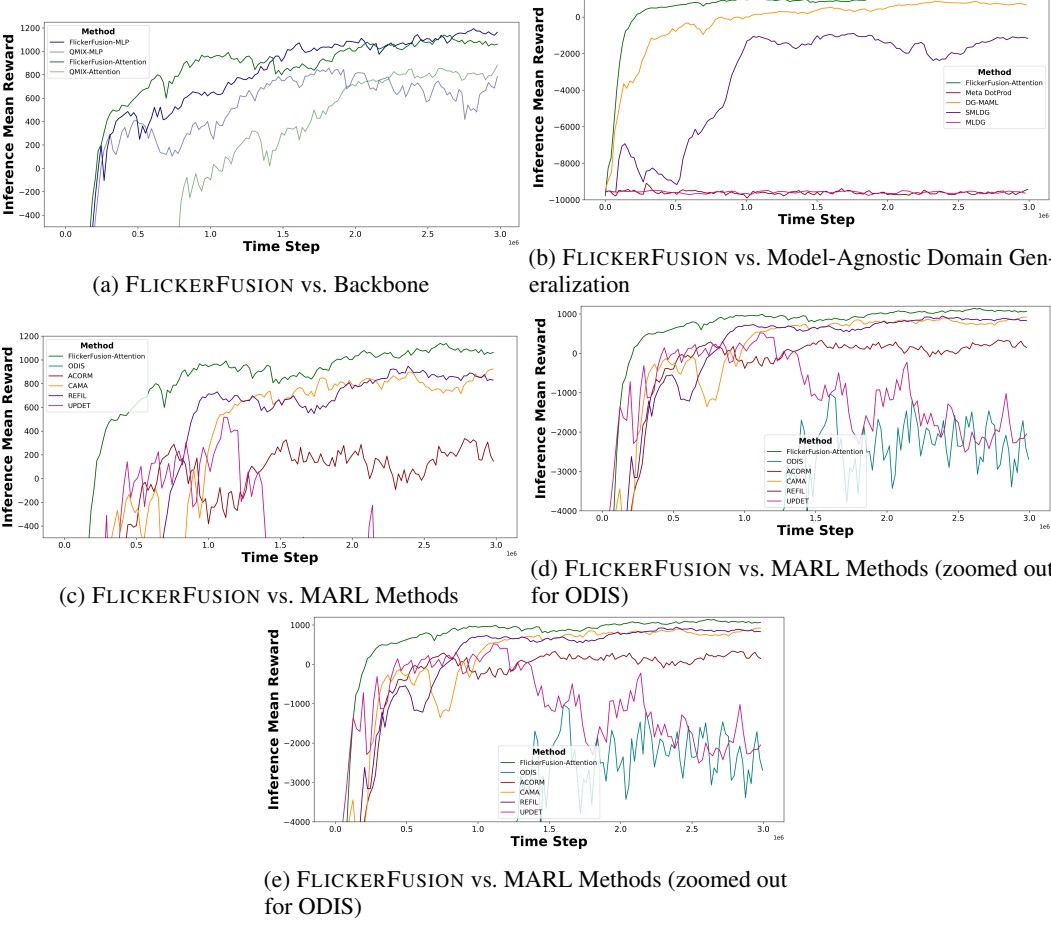

(a) FLICKERFUSION vs. Backbone

(b) FLICKERFUSION vs. Model-Agnostic Domain Generalization

(c) FLICKERFUSION vs. MARL Methods

(d) FLICKERFUSION vs. MARL Methods (zoomed out for ODIS)

(e) FLICKERFUSION vs. MARL Methods (zoomed out for ODIS)

Figure 7: Empirical results on Repel (OOD2)

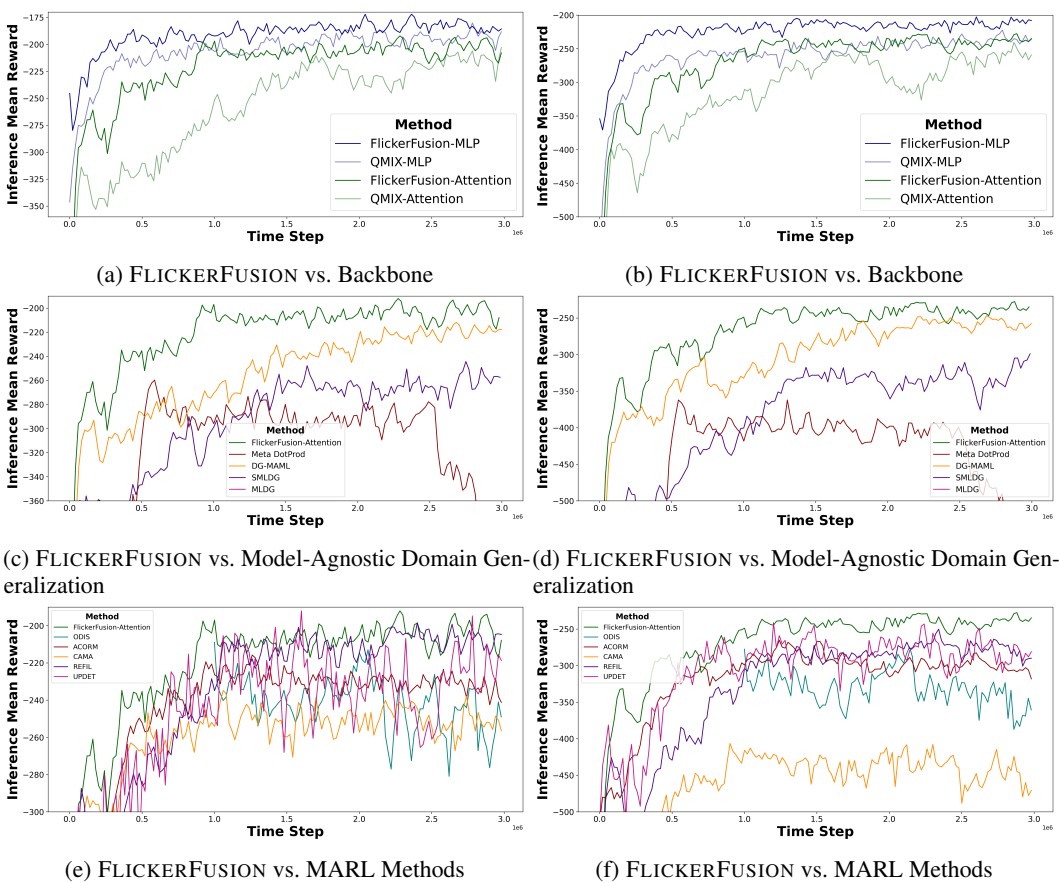

(a) FLICKERFUSION vs. Backbone

(b) FLICKERFUSION vs. Backbone

(c) FLICKERFUSION vs. Model-Agnostic Domain Generalization

(d) FLICKERFUSION vs. Model-Agnostic Domain Generalization

(e) FLICKERFUSION vs. MARL Methods

(f) FLICKERFUSION vs. MARL Methods

Figure 8: Empirical results on Spread (left: OOD1, right: OOD2)

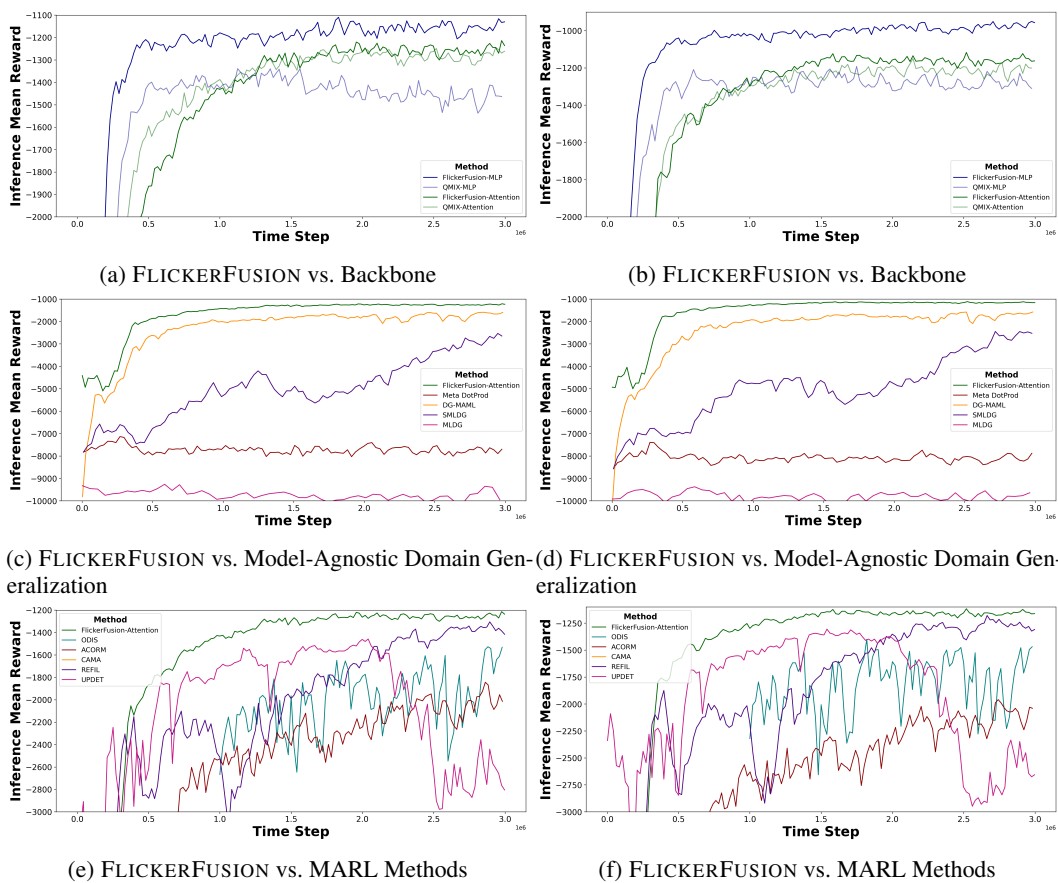

(a) FLICKERFUSION vs. Backbone

(b) FLICKERFUSION vs. Backbone

(c) FLICKERFUSION vs. Model-Agnostic Domain Generalization

(d) FLICKERFUSION vs. Model-Agnostic Domain Generalization

(e) FLICKERFUSION vs. MARL Methods

(f) FLICKERFUSION vs. MARL Methods

Figure 9: Empirical results on Guard (left: OOD1, right: OOD2)

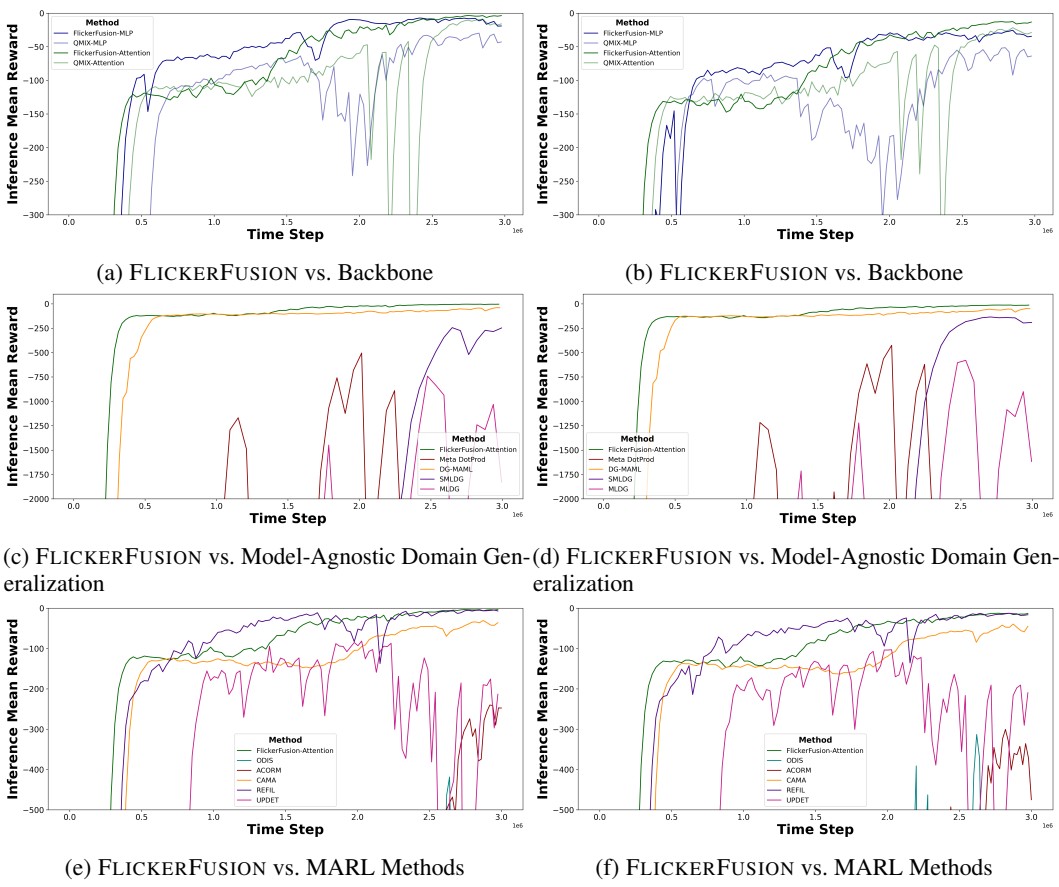

(a) FLICKERFUSION vs. Backbone

(b) FLICKERFUSION vs. Backbone

(c) FLICKERFUSION vs. Model-Agnostic Domain Generalization

(d) FLICKERFUSION vs. Model-Agnostic Domain Generalization

(e) FLICKERFUSION vs. MARL Methods

(f) FLICKERFUSION vs. MARL Methods

Figure 10: Empirical results on Tag (left: OOD1, right: OOD2)

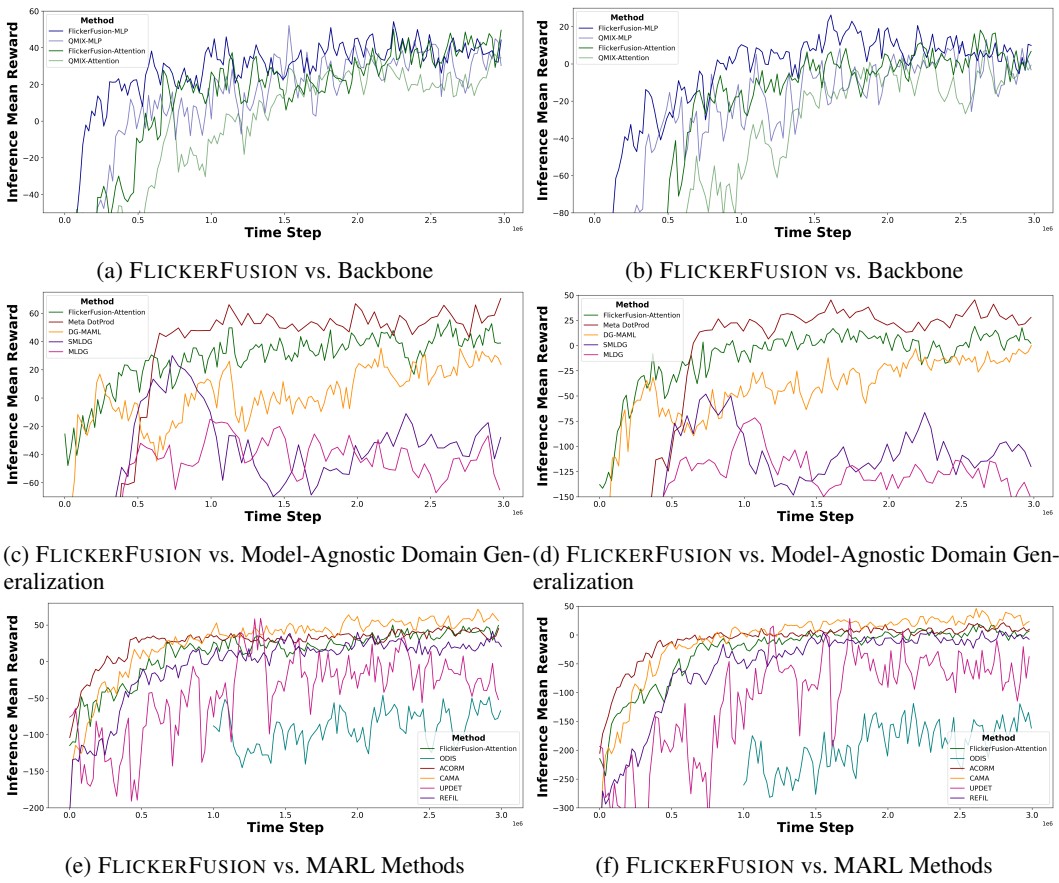

(a) FLICKERFUSION vs. Backbone

(b) FLICKERFUSION vs. Backbone

(c) FLICKERFUSION vs. Model-Agnostic Domain Generalization

(d) FLICKERFUSION vs. Model-Agnostic Domain Generalization

(e) FLICKERFUSION vs. MARL Methods

(f) FLICKERFUSION vs. MARL Methods

Figure 11: Empirical results on Adversary (left: OOD1, right: OOD2)

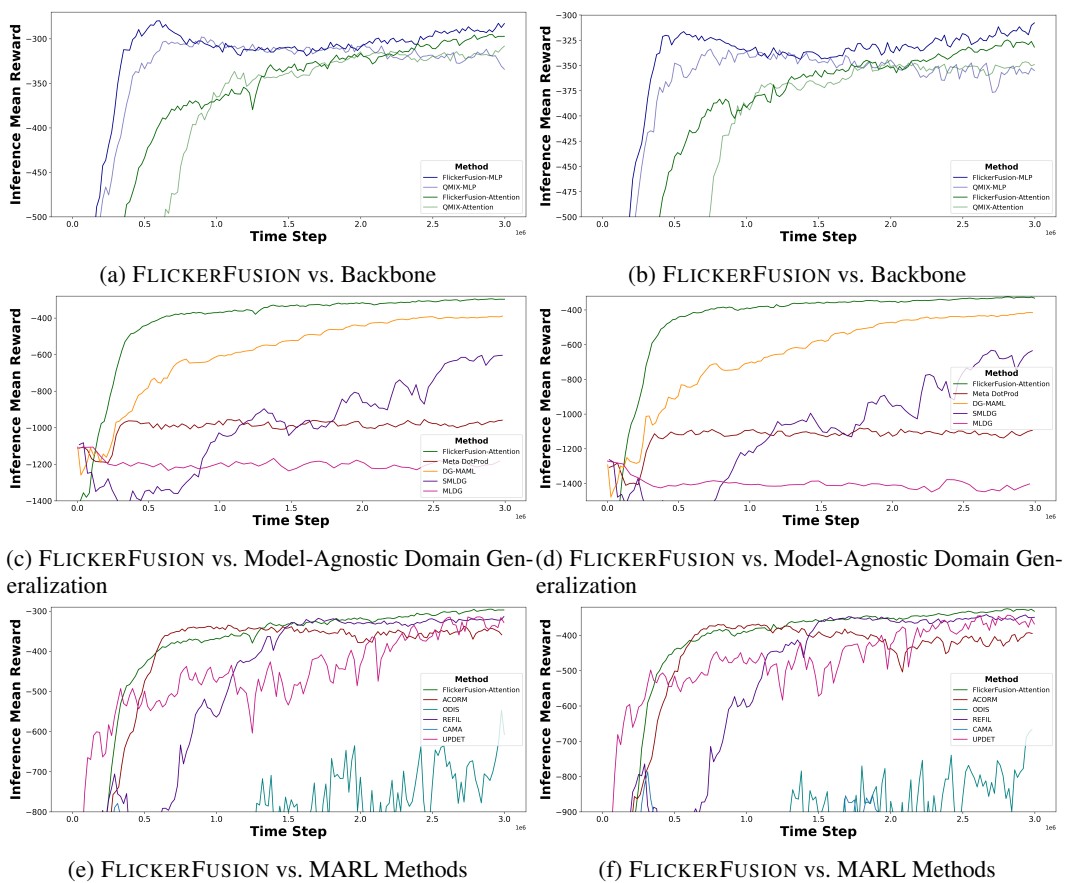

(a) FLICKERFUSION vs. Backbone

(b) FLICKERFUSION vs. Backbone

(c) FLICKERFUSION vs. Model-Agnostic Domain Generalization

(d) FLICKERFUSION vs. Model-Agnostic Domain Generalization

(e) FLICKERFUSION vs. MARL Methods

(f) FLICKERFUSION vs. MARL Methods

Figure 12: Empirical results on Hunt (left: OOD1, right: OOD2)

# F    AGENT-WISE POLICY HETEROGENEITY

Diversity across rows in Figure 13 highlight agent-wise policy heterogeneity. We visualize the attention matrix of QMIX-Attention and FLICKERFUSION-Attention during inference. For consistent visualization, we snapshot the matrices at the middle of the episode ($t_{max}/2$). Resolving agent-wise homogeneity enhances the aggregate strategic expressivity of the cooperative system. This can be qualitatively validated on our site's demo rendering videos. We visualize all attention matrices for the remaining benchmarks in (Figure 14 to 22).

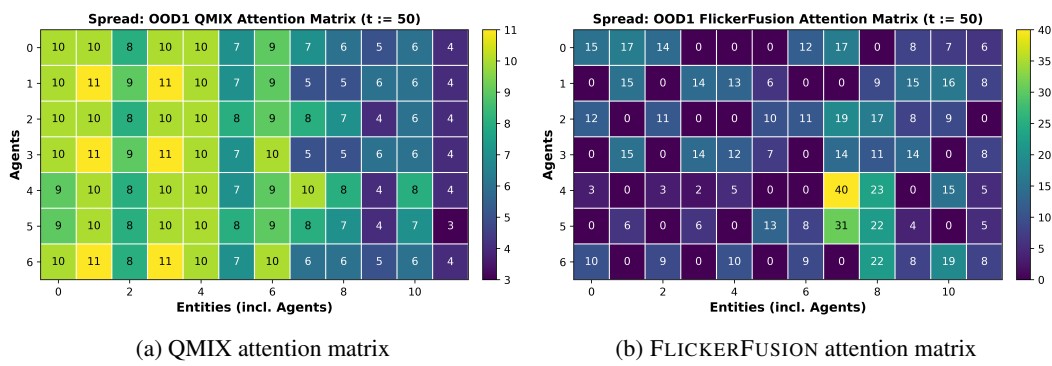

(a) QMIX attention matrix            (b) FLICKERFUSION attention matrix

Figure 13: Agent-wise policy heterogeneity significantly improved in Spread (OOD1)

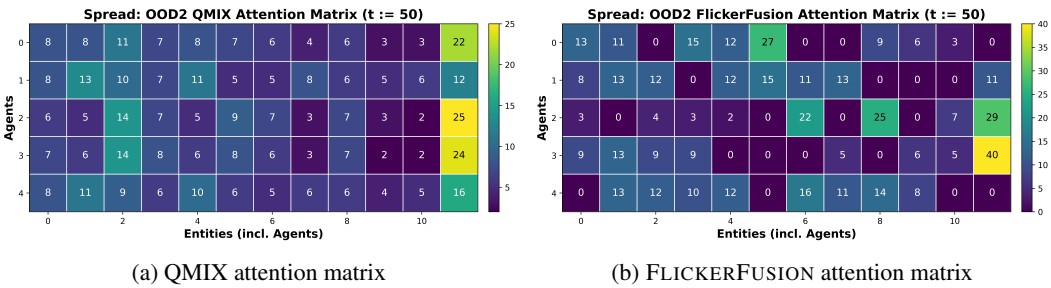

(a) QMIX attention matrix            (b) FLICKERFUSION attention matrix

Figure 14: Agent-wise policy heterogeneity improved in Spread (OOD2)

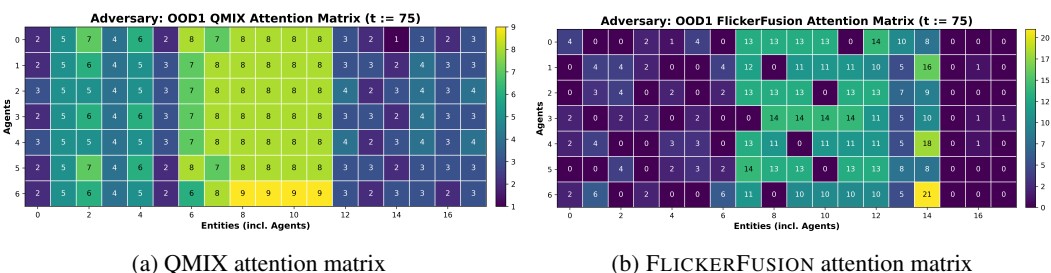

(a) QMIX attention matrix            (b) FLICKERFUSION attention matrix

Figure 15: Agent-wise policy heterogeneity improved in Adversary (OOD1)

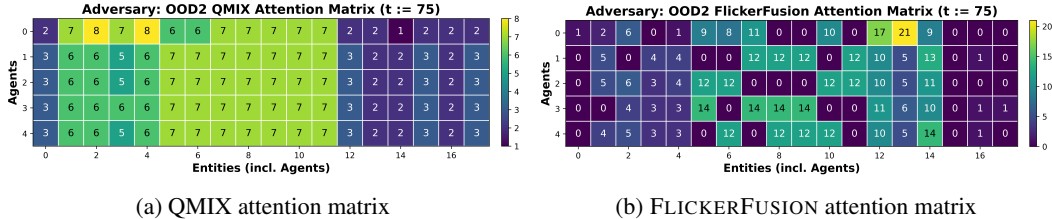

(a) QMIX attention matrix

(b) FLICKERFUSION attention matrix

Figure 16: Agent-wise policy heterogeneity improved in Adversary (OOD2)

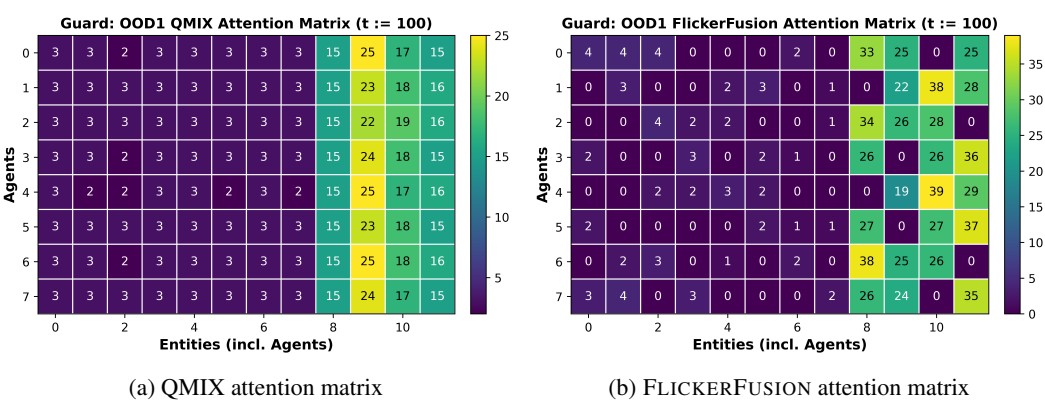

(a) QMIX attention matrix

(b) FLICKERFUSION attention matrix

Figure 17: Agent-wise policy heterogeneity improved in Guard (OOD1)

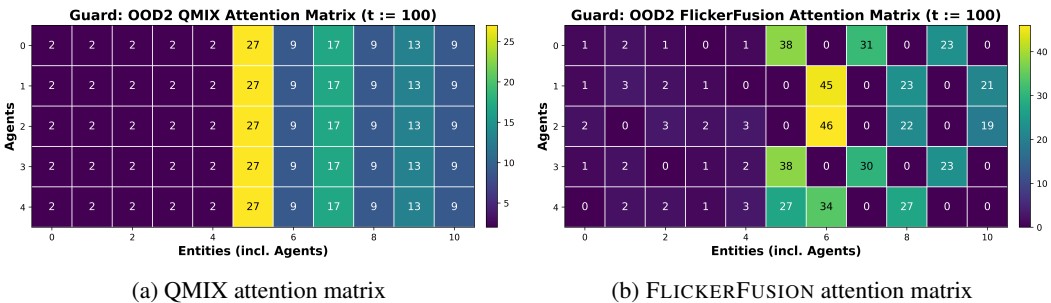

(a) QMIX attention matrix

(b) FLICKERFUSION attention matrix

Figure 18: Agent-wise policy heterogeneity improved in Guard (OOD2)

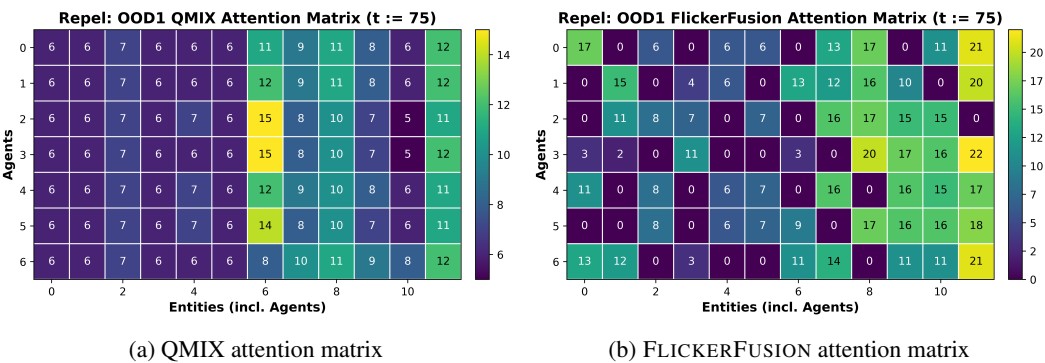

(a) QMIX attention matrix

(b) FLICKERFUSION attention matrix

Figure 19: Agent-wise policy heterogeneity improved in Repel (OOD1)

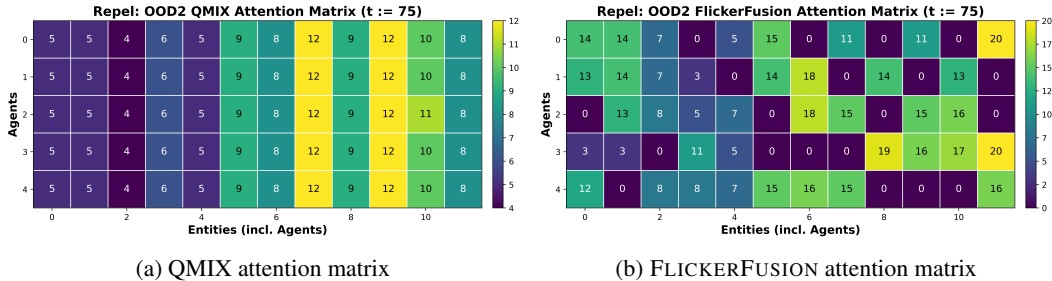

(a) QMIX attention matrix

(b) FLICKERFUSION attention matrix

Figure 20: Agent-wise policy heterogeneity improved in Repel (OOD2)

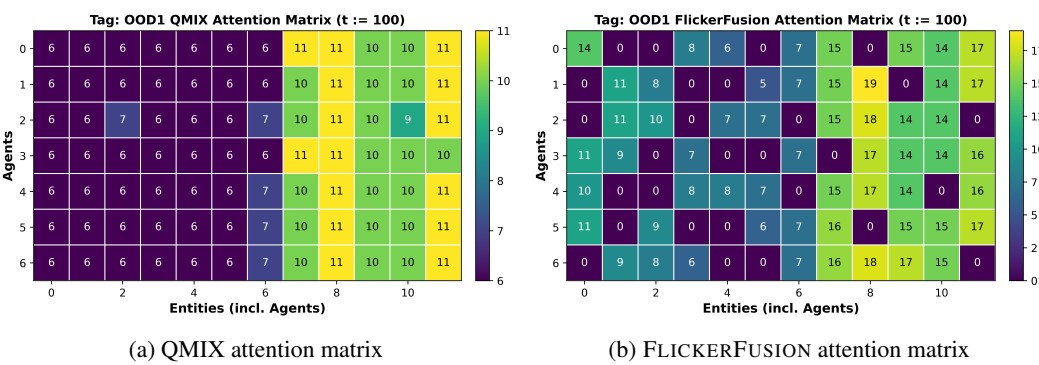

(a) QMIX attention matrix

(b) FLICKERFUSION attention matrix

Figure 21: Agent-wise policy heterogeneity improved in Tag (OOD1)

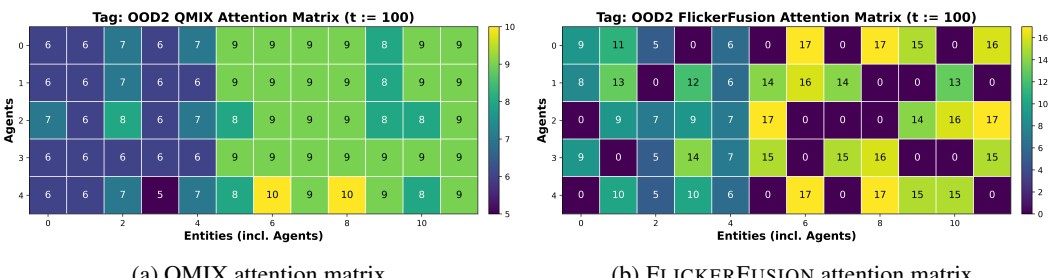

(a) QMIX attention matrix

(b) FLICKERFUSION attention matrix

Figure 22: Agent-wise policy heterogeneity improved in Tag (OOD2)

## G    RELATIONSHIP BETWEEN REWARD AND UNCERTAINTY

See Fig. 23 and 24.

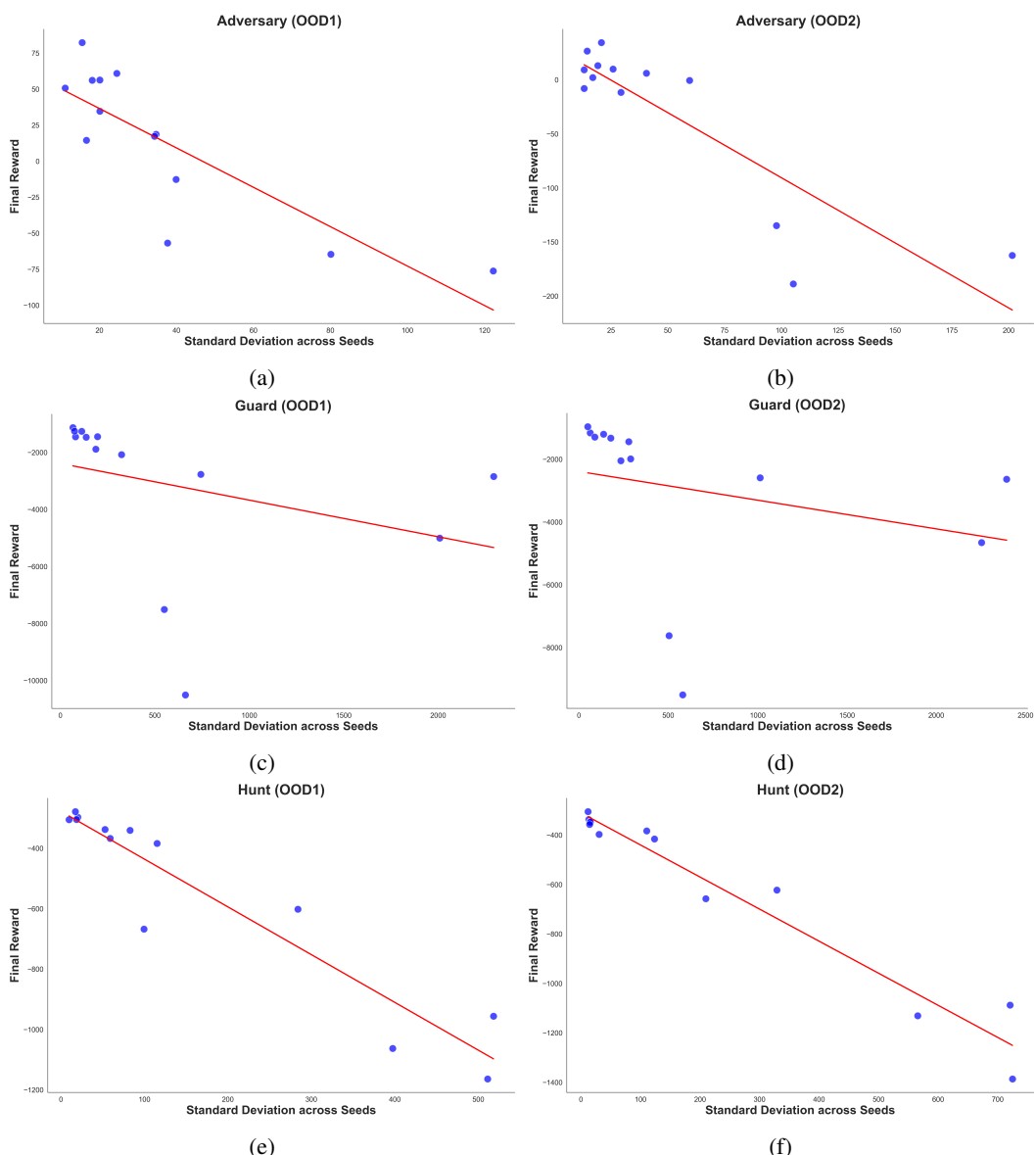

Figure 23: Negative relationship between final reward and seed-wise uncertainty (left: OOD1, right: OOD2). Blue dots are results for each method. Red curve is fitted.

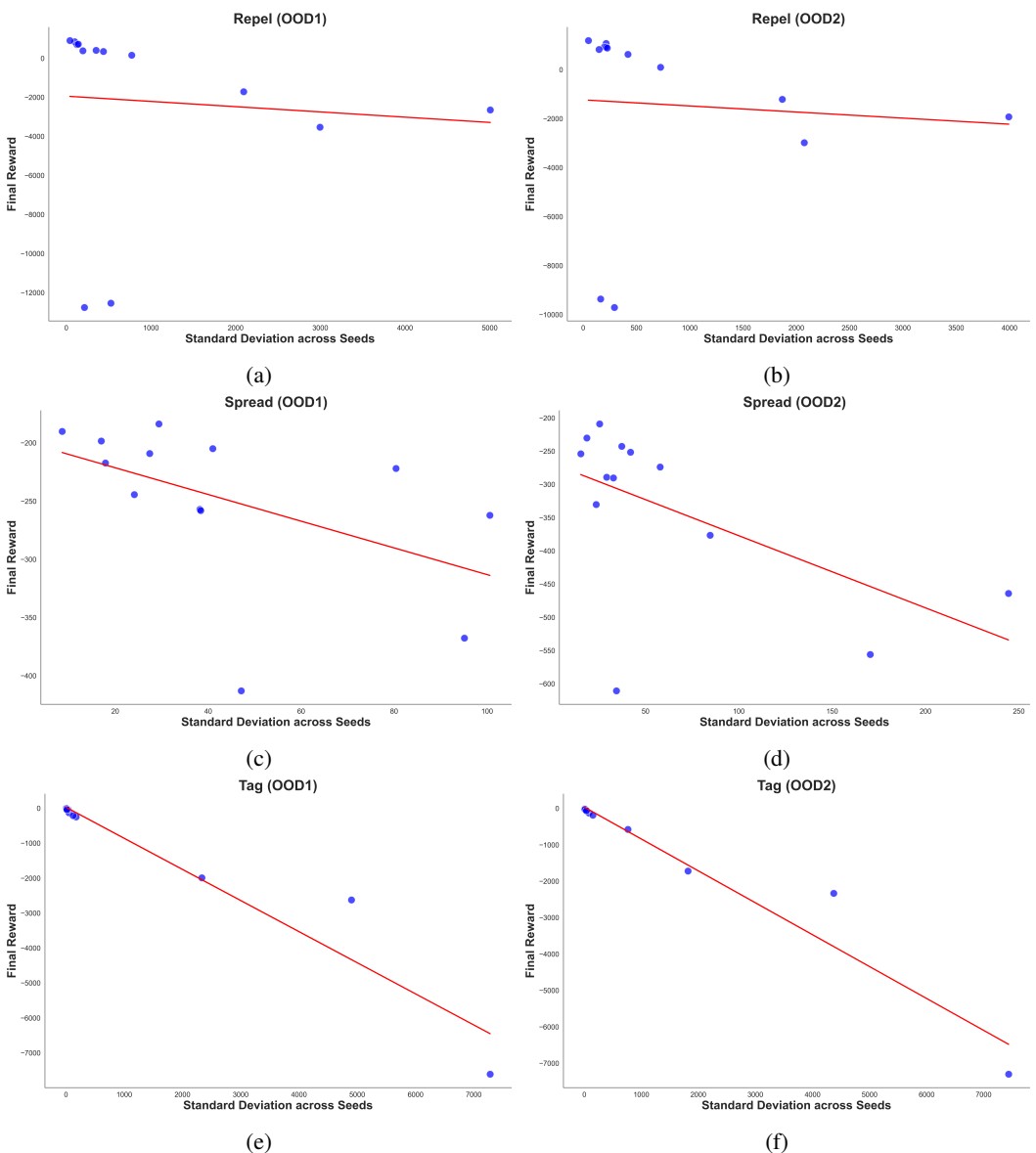

Figure 24: Negative relationship between final reward and seed-wise uncertainty (left: OOD1, right: OOD2). Blue dots are results for each method. Red curve is fitted.

