# OpenReview forum: "FlickerFusion: Intra-trajectory Domain Generalizing Multi-agent Reinforcement Learning"
_ICLR.cc/2025/Conference — ICLR 2025 Poster_

### Official Review · Reviewer_imDX · 2024-11-02

**Soundness:** 3
**Presentation:** 3
**Contribution:** 3
**Rating:** 6
**Confidence:** 4

**Summary:**

The paper introduces FLICKERFUSION,  a novel method aimed at enhancing out-of-domain (OOD) generalization in Multi-Agent Reinforcement Learning (MARL) environments where the composition of entities changes dynamically within a trajectory. The authors identify two significant challenges: intra-trajectory dynamic entity addition and zero-shot OOD generalization, which existing methods fail to handle effectively. FLICKERFUSION utilizes stochastic input-space dropout to maintain observation space consistency without additional parameters, achieving superior performance across a newly developed benchmark suite, MPEV2. Empirical results show that FLICKERFUSION outperforms existing approaches in both inference reward and uncertainty reduction.

**Strengths:**

1.The paper addresses a critical yet under-explored problem in MARL, specifically the intra-trajectory dynamic entity composition and its impact on generalization. The proposed approach, which uses entity dropout and stochastic fusion, is original and impactful.

2.The introduction of the MPEV2 benchmark enhances the paper's contribution, allowing a rigorous assessment of the proposed method. The results across various environments are convincing and demonstrate clear performance improvements.

3.The code and benchmarks are made publicly available, supporting reproducibility and further research in this domain.

4.This paper includes many baselines to  validate its efficiency.

**Weaknesses:**

1.This paper addresses some open problems in MARL. However, it lacks discussion on several related areas, such as multi-agent policy transfer [1] and sudden policy changes [2]. Including a more comprehensive analysis of these works could strengthen the paper.

2.The definition of OOD is unclear and too simple.

3.The use of the 'Flicker Fusion Phenomena' in this study is intriguing. However, the random masking approach might result in the loss of team semantic information. Alternative methods, such as learning to group[3] or applying the widely used mean-field theory in MARL[4], could potentially offer more advantages. Could the authors elaborate on the strengths and weaknesses of their approach compared to these methods?

4.Could this paper be further enhanced by incorporating Large Language Models (LLMs) like ChatGPT[5]?

Ref

[1] Qin R, Chen F, Wang T, et al. Multi-Agent Policy Transfer via Task Relationship Modeling[J]. arXiv e-prints, 2022: arXiv: 2203.04482.

[2] Zhang Z, Yuan L, Li L, et al. Fast teammate adaptation in the presence of sudden policy change[C]//Uncertainty in Artificial Intelligence. PMLR, 2023: 2465-2476.

[3] Sheng, Junjie, et al. "Learning structured communication for multi-agent reinforcement learning." Autonomous Agents and Multi-Agent Systems 36.2 (2022): 50.

[4] Yang Y, Luo R, Li M, et al. Mean field multi-agent reinforcement learning[C]//International conference on machine learning. PMLR, 2018: 5571-5580.

[5] Sun C, Huang S, Pompili D. LLM-based Multi-Agent Reinforcement Learning: Current and Future Directions[J]. arXiv preprint arXiv:2405.11106, 2024.

**Questions:**

1.How does FLICKERFUSION perform in environments with extremely high or low densities of entity changes? Is the dropout strategy adaptable enough to handle these scenarios effectively?

2.Could this method be extended to other benchmarks, such as SMAC, MAgent, or CityFlow?

Ref

[1] Yuan L, Zhang Z, Li L, et al. A survey of progress on cooperative multi-agent reinforcement learning in open environment[J]. arXiv preprint arXiv:2312.01058, 2023.

---

> ### Author Response · Authors · 2024-11-19
> **Response to Reviewer imDX (Part I)**
>
> Thank you for your meticulous feedback and sharing your time contributing to the community. We address your specific concerns **[C]** and questions **[Q]** in detail below.
>
>
>
>
> ### **[C1] Two Additional Related Works**
>
>
>
> We have included these two additional works in **Sec. 1** and **Sec. 5**, ensuring that contributors to the community are appropriately accredited. For clarity, we will describe how these two approaches are *related but different* to ours in **(1)** for [1], and **(2)** for [7].
>
>
>
> **(1)** First, [1] is a valuable contribution where they leverage multiple tasks to improve generalization, but has no *intra-trajectory* dynamic entity composition. This work is an important step towards large scale pre-training for MARL. Nevertheless, this work primarily focused on *knowledge transfer* using multiple tasks [1]. While we acknowledge that [1] has numerous objectives with *generalization only being one of them*, we would like to share some strengths of our work compared to [1], in terms of generalization:
>
> 1.  Our evaluation is more concretely out-of-domain (OOD). Consider **Tab. 4** and **5** in [1] where MPE-based evaluation is used. In the case of **Tab. 4** in [1], their unseen tasks are interpolations of the seen tasks. Concretely, the source task is 2, 4, 7, 9 agents while the evaluation setting is 3, 5, 6, 8 agents. While **Tab. 5** in [1] provides two extrapolation cases, our evaluation setting as visualized in **Fig. 3 (a)** is *wholly extrapolation*, and therefore, more evidently OOD. It is well known that deep learning has strong interpolation abilities. We directly quote: “...deep neural networks are excellent tools in interpolating…” [2] and “...generalization exhibited by contemporary neural network algorithms is largely limited to interpolation…” [3].
> 2.  Our evaluation baselines are more comprehensive. [1] has *two baselines* [4, 5], both published in **2021**. Alternatively, we have *11 baselines*, all from top AI/ML venues ranging from **2018** to **2024**. Our view is that CAMA [6] is the most relevant one, published at ICML 2023, which we very clearly beat. We acknowledge that [1] has other baselines for other (non-zero-shot generalization) objectives, but we focus on the zero-shot generalization case as it is similar to our problem setting.
> 3.  Our method always leads to state-of-the-art uncertainty reduction—i.e., smaller standard deviation against baselines (see **Fig. 5 top-left**). However, this is not the case for [1]. As seen in **Tab. 4** and **5** in [1], in the unseen task columns, the average variance (as reported in the table) of [1] is *greater* than baseline(s). Moreover, notice that column 10, 15 (**extrapolated** unseen tasks) in **Tab. 2** below, results in an *extraordinary jump in instability* for [1]. We highlight this with ${\color{red}red}$. Here are the tables:
>
> **Table 1: Variance Comparison (from Tab. 4 in [1])**
> | Method    | 3    | 5    | 6    | 8    | Mean  | Median |
> |-----------|------|------|------|------|-------|--------|
> | [1]    | 0.01 | 0.12 | 0.10 | 0.09 | 0.080 | 0.095  |
> | UPDeT-b   | 0.16 | 0.02 | 0.00 | 0.02 | 0.050 | 0.020  |
> | UPDeT-m   | 0.08 | 0.01 | 0.00 | 0.01 | **0.025** | **0.010** |
> | REFIL     | 0.03 | 0.06 | 0.05 | 0.00 | 0.035 | 0.040  |
>
> **Caption:** Comparing variance of MATTAR vs. baselines on unseen tasks. Best value in mean and median is **bolded**.
>
> **Table 2: Variance Comparison (from Tab. 5 in [1])**
> | Method    | 3    | 5    | 10   | 15   | Mean  | Median |
> |-----------|------|------|------|------|-------|--------|
> | [1]    | 0.00 | 0.00 | ${\color{red}0.12}$ | ${\color{red}0.22}$ | 0.085 | 0.060  |
> | UPDeT-b   | 0.36 | 0.27 | 0.03 | 0.06 | 0.180 | 0.165  |
> | UPDeT-m   | 0.10 | 0.07 | 0.01 | 0.02 | **0.050** | **0.045** |
> | REFIL     | 1.00 | 0.00 | 0.14 | 0.14 | 0.320 | 0.140  |
>
> **Caption:** Comparing variance of MATTAR vs. baselines on unseen tasks. Best value in mean and median is **bolded**.
>
> **(2)** Second, we describe how [7] differs drastically from our work. [7] is focused on a *highly unique  scenario* where a teammate's policy suddenly changes. Additionally, they include collaborating with uncontrollable teammates, making it completely different from ours and the *11 baselines* we include. Moreover, the parameterized agents that they control do not have intra-trajectory changing entity composition.

---

> ### Author Response · Authors · 2024-11-19
> **Response to Reviewer imDX (Part II)**
>
> ### **[C2] OOD Definition**
>
>
>
> We believe that our OOD definition has been clearly stated in the paper. First, in **Sec. 3.1**, we presented a rigorous mathematical definition in **Eq. (3)**. This definition is the foundation to the entire **Sec. 3**. which is our methods section. To further ensure that the readers have an intuitive understanding of the definition, we also have a visualization of the definition in **Fig. 3 (a)**. It is even color-coded for an intuitive linkage with **Fig. 3 (b)** and **(c)**. Please let us know if you believe we can clarify it even further. Lastly, while our definition may be fairly straight-forward, it is a common and critical OOD scenario for MARL deployment (see **Sec. 1**, [6], [17]). We do not believe that its simplicity negatively affects its salience.
>
> ### **[C3] Flicker Fusion Phenomena, and Drawback of Random Dropout**
>
>
>
> While some information is lost when applying our method, the trade-off we describe below, is significantly swayed to our favor (↑ reward, ↓ uncertainty). We empirically prove this by beating *11 baselines* across *12 benchmarks*. We discuss the details on how this is achieved in **(1)**, then address the two other potential solutions you provide in **(2)**.
>
>
>
> **(1)** As detailed in **Sec. 3.1**, the primary challenge of zero-shot domain generalization in MARL is being forced to initialize more parameters, from $\theta_Q^{D_I}$ to $\theta_Q^{D_O} := \theta_Q^{D_I} \oplus \theta_Q'$ to accommodate an increased observation vector (**page 4**). This occurs under both MLP and Attention backbones. As described in **Sec. 3.2**, all prior works attempt to inject inductive biases within the newly initialized parameters, $\theta_Q'$. Orthogonally, our approach stochastically drops out observations at each timestep.
>
>
>
> In short, there is a trade-off between “test-time new parameter initialization” and “test-time stochastic observation”, and we empirically show that the latter approach is far better relative to initializing new parameters under OOD inference. Your intuition is correct that a naïve dropout may result in sub-optimal results (**Sec. 3.2**). However, the design of FlickerFusion allows us to minimize the information loss from the dropouts. This is achieved by allowing FlickerFusion to adaptively decide **“How Many to Drop?” (Sec. 3.2)** and “**Which to Drop?” (Sec. 3.2, 3.3)** on-the-fly, if it detects OOD. In essence, **“How Many to Drop?” (Sec. 3.2)** is the domain-aware part—where  it only drops the *minimum entities required* to emulate being in-domain (see **Eq. (6)** and **Alg. 2 line 6**). It is also trained such that it learns to perform well under random entity dropouts (**Alg. 1 line 6**). I.e, as mentioned in-text: “The training is done similarly to “prepare” for the entity dropouts that the agents will experience at test time”. As we have open-sourced all *12 benchmarks*, *11 baselines*, our *method*, and *model weights* at the time of submission, we are fully transparent for Reviewers, AC, and PC to try it out. You can also view the demo visualizations on our anonymized project site (flickerfusion305.github.io), which qualitatively illustrates that the information loss does not meaningfully deter performance.
>
>
>
> **(2)** We now respond to other potential solutions [8, 9] that you have shared. [8]’s problem setting is different from ours and *all 11 baselines* that we evaluate on, as it is *not* Centralized Training and Decentralized Execution (CTDE) [10, 11]. Specifically, [8] is not decentralized execution, making it inapplicable to many real-world scenarios that require decentralized execution, such as search and rescue and dynamic combat, as discussed in our **Sec. 1**. We additionally remark that centralized execution embodies a significant network cost burden that must be robust at all times for stable functioning. This can be embraced in certain scenarios but is *not as generally plausible* as CTDE. Partly owing to this, CTDE is the mainstream problem setting in MARL research [4, 5, 6, 12, 13], which we follow.
>
>
>
> [9]’s approach was one of the first (published 2018) attempts at incorporating (mean-field) game-theoretic ideas into MARL [5]. It focuses on alleviating the curse of dimensionality that arises when training a large number of agents, which is fundamentally different from our problem setting, which focuses on *dynamic entity composition*, and *domain generalization*. Instead, we compare our methodology with more relevant works. As mentioned in **Sec. 5**, we have included all possible relevant baselines. Among these, five of them are most relevant, color-coded yellow in **Tab. 1**. All five have been accepted as claiming state-of-the-art at $\langle$ ICML, ICLR, NeurIPS $\rangle$, but none include [9] as a baseline, highlighting the difference in problem setting. Note, all these works have come after [9] vis-à-vis chronology.

---

> ### Author Response · Authors · 2024-11-19
> **Response to Reviewer imDX (Part III)**
>
> ### **[C4] Can LLMs Improve This?**
>
>
>
> LLMs have the *potential* to be inductive bias providers [14], yet, they come with *clear* drawbacks in terms of scalability in real-life deployment. As presented in [14], the *smallest* LLM model that has been reasonably applied to embodied applications is **8B** in size. Considering that the *de facto* CTDE MARL framework [10, 11] needs decentral execution, such LLMs need to be integrated onto the edge-device per entity, e.g., a drone. In contrast, our models are **<500K** parameter size, as are all other CTDE MARL methods [4, 5, 6, 10, 11, 12, 13]. This is an *indispensable advantage* in the CTDE setting. Nevertheless, we acknowledge that this can be interesting future work, and we believe that advancements in edge-device LLMs and CTDE MARL can mutually benefit from each other.
>
> ### **[Q1] Dramatic Entity Change**
>
>
>
> It is well known that the performance of the neural network deteriorates as the degree of out-of-domain (OOD) increases, i.e., more dis-similar to the in-domain [2, 3]. Thus, we believe that such an extreme situation of dramatic entity change (as the reviewer has suggested) is intrinsically difficult. The objective of zero-shot domain generalization is *making best* of the challenging situation at test-time [6, 15]. We do not claim that our method achieves perfect optimality, but we still show that it results in *improved robustness* (↑ reward, ↓ uncertainty) against *all 11 baselines*. To demonstrate this, we have ensured that the 12 evaluations are done 100% on OOD scenarios, with *extrapolated* (not interpolated) entity composition (see **Fig. 3 (a)**, **Tab. 1**).
>
>
>
> Let us elaborate by recalling some of the exact numbers from the Guard environment (see our **App. B.4**). For simplicity, we discuss one entity type: the parameterized agent. 1 to 4 agents are seen by the learner during training, and in inference, it is evaluated on 5 to 8 agents. This means that our method displays strong performance even as the maximum number of agents has jumped *two-fold* ($4 \rightarrow 8$). As your intuition suggests, we believe that our method (but *even more so the baselines*) will struggle as we continue extrapolating further to, e.g., $4 \rightarrow 20$.
>
>
>
> ### **[Q2] Extension to Other Benchmarks**
>
>
>
> Yes, we see no reason why it would not transfer to other environments. As visualized in **Fig. 2**, our method is a *universal plug-in* method to any backbone (MLP or Attention-based). We believe that the simplicity of our method is a meaningful advantage; it can be easily paired with any existing architecture and method.
>
>
>
> We would also like to direct you to consider a previous ICLR paper [16] which builds upon MPE to demonstrate the effectiveness of their approach (https://sites.google.com/view/epciclr2020). MPE, SMAC, MAgent, and CityFlow are abstractions (simplifications) of real-world scenarios, allowing researchers to efficiently improve their methods in a *tractable* setting. We believe that having a diverse range of potential environments play an integral and *complementary* role in advancing the state-of-the-art.
>
>
>
> ### **Reference**
> [1] Qin et al. "Multi-agent policy transfer via task relationship modeling." SCIS (Journal, 2024).
>
> [2] Ziyin et al. "Neural networks fail to learn periodic functions and how to fix it." NeurIPS 2020.
>
> [3] Webb et al. "Learning representations that support extrapolation." ICML 2020.
>
> [4] Iqbal et al. “Randomized Entity-wise Factorization for Multi-Agent Reinforcement Learning.” ICML 2021.
>
> [5] Hu et al. “UPDeT: Universal Multi-agent RL via Policy Decoupling with Transformers.” ICLR 2021.
>
> [6] Shao et al. “Complementary Attention for Multi-Agent Reinforcement Learning.” ICML 2023.
>
> [7] Zhang et al. "Fast teammate adaptation in the presence of sudden policy change." UAI 2023.
>
> [8] Sheng et al. "Learning structured communication for multi-agent reinforcement learning." AAMAS 2022.
>
> [9] Yang et al. "Mean field multi-agent reinforcement learning." ICML 2018.
>
> [10] Tabish et al. “QMIX: Monotonic Value Function Factorisation for Deep Multi-Agent Reinforcement Learning.” ICML 2018.
>
> [11] Jakob et al. “Stabilising Experience Replay for Deep Multi-Agent Reinforcement Learning.” ICML 2017.
>
> [12] Hu et al. “Attention Guided Contrastive Role Representations for Multi-agent Reinforcement Learning.” ICLR 2024.
>
> [13] Zhang et al. “Discovering Generalizable Multi-agent Coordination Skills from Multi-task Offline Data.” ICLR 2023.
>
> [14] Sun et al. "LLM-based Multi-Agent Reinforcement Learning: Current and Future Directions." arXiv:2405.11106 (2024).
>
> [15] Yuan et al. "Rl-vigen: A reinforcement learning benchmark for visual generalization." NeurIPS 2024.
>
> [16] Long et al. "Evolutionary Population Curriculum for Scaling Multi-Agent Reinforcement Learning." ICLR 2020.
>
> [17] Zhang et al. "Discovering generalizable multi-agent coordination skills from multi-task offline data." ICLR 2023.

---

> ### Comment · Reviewer_imDX · 2024-11-20
> **Thank your for detailed response**
>
> Thank you for the detailed response, I have adjusted my score accordingly.
>
> Additionally, I believe this paper focuses on MARL in open-environment settings and suggest including a comparison with other methods[1].
>
>
> [1] A survey of progress on cooperative multi-agent reinforcement learning in open environment[J]. arXiv preprint arXiv:2312.01058, 2023.

---

> > ### Author Response · Authors · 2024-11-24
> > **Thank you**
> >
> > Thank you for your time and effort. We are carefully reading the survey paper you have shared with us.
> >
> > Best, Authors.

---

### Official Review · Reviewer_wmnB · 2024-11-03

**Soundness:** 3
**Presentation:** 3
**Contribution:** 4
**Rating:** 8
**Confidence:** 3

**Summary:**

This paper addresses multi-agent reinforcement learning (MARL) with a dynamic number of entities. Unlike existing methods that introduce extra parameters, the proposed novel method, FLICKERFUSION, functions as an augmentation technique that selectively drops entities to ensure the observation space aligns with the policy network, regardless of changes in the number of entities. The experiments and ablation studies demonstrate the superior performance of the proposed method and clarify its operational mechanisms.

**Strengths:**

1. The issue of zero-shot out-of-domain (OOD) generalization in MARL is significant in bridging the gap between laboratory MARL research and real-world deployment.
2. The concept of FLICKERFUSION, which involves dropping out entities and recovering lost information, is novel and inspiring.
3. Although the paper uses only one series of benchmarks, MPEV2, it includes several different scenarios and baselines to demonstrate the robust performance of the proposed method.
4. The presentation is clear, and the details are effectively conveyed in the text and the Appendix.

**Weaknesses:**

1. The conceptualization of the paper appears to be excessive. For example, "intra-trajectory" is discussed in detail in the introduction section but is not mentioned in the methods section, which may confuse the audience. Additionally, some key points and implementations of the proposed method are not conveyed in the introduction. For instance, the technique of dropout is missing from both the abstract and the introduction.
2. The paper seems to overlook situations where there are different types of entities with varying lengths and/or compositions of observation space. In such cases, the stochastic dropout approach may not be effective.

**Questions:**

1. How would the proposed method operate in situations where there are different types of entities with varying lengths and/or compositions of observation space?
2. In Figure 5, I suggest that the authors use a different line type for the proposed method to clearly distinguish FLICKERFUSION from the baselines.

---

> ### Author Response · Authors · 2024-11-19
> **Response to Reviewer wmnB**
>
> We are grateful for your careful analysis and thoughtful feedback. We address your specific concerns **[C]** and questions **[Q]** in detail below to further clarify and help the readers’ understanding.
>
>
>
> ### **[C1] Intra-trajectory in Methods Section, and Mention Dropout in Abstract and Introduction**
>
>
>
> We enhance our paper by addressing your first concern on our paper being excessively conceptualized below, and second concern on lack of discussions regarding dropout in the **global response**.
>
>
>
> We acknowledge that the lack of intra-trajectory mentioned in **Sec. 3** can be puzzling as we have thoroughly discussed intra-trajectory in **Sec. 1**. In response to your feedback, we have updated the text to guide the readers on this salient remark. In **Sec. 3.3**:
>
> -   “... added benefit that any intra-trajectory dynamics changes can be immediately responded to on-the-fly.”
> -   “As seen by the ${\color{DarkGreen}\textbf{green}}$, executing the algorithm at every $t$ ensures smooth transition to any intra-trajectory OOD dynamics.”
>
>
> Moreover, we have made a minor update to the image at **Fig. 4**, emphasizing this remark.
>
>
>
> ### **[C2, Q1] Differing Entity Types, Observation Sizes, and Composition**
>
>
>
> Indeed, as you pointed out, there can be different entity types, where each entity type has a unique observation size, and therefore, dynamic composition. We are glad to share that our method is *fully compatible* under these dynamic scenarios. Using the notations in our paper we explain how this is possible in **(1)**. For clarity, we further provide a toy example in **(2)**.
>
>
>
> **(1)** First, we would like to direct you to **Sec. 2 Preliminary**. Following the *de facto* observation composition of the MARL literature [2, 3], $s^e := f^e \oplus \phi^e$ where $f^e \in \mathbb{R}^{d_{\phi^e}}$ is the feature vector (e.g., velocity, location), and for each entity $e \in \mathcal{E}$, $\phi^e \in \Phi$ is its type vector. Here, unlike the traditional neuron dropout [1], which drops out scalar values within the vectors, we are dropping out in *units of vectors*; concretely, we are dropping state representation $s^e$. Therefore, even as the size of $s^e$ differs for each $e$, there is no problem dropping out across *different entity types* ($e$). As further clarified in **Sec. 3.2**, “This is done by dropping $\max(0, N_\ell^{inf} - N_\ell^{train})$ entities for each type $e_\ell$” and “each agent independently and uniformly samples $\Delta[\ell]$ number of entities of type $\ell$ to drop”. I.e., the drop outs are contingent on entity type, $e$.
>
>
>
> **(2)** Here is a concrete toy example using the same logic proposed in **Sec. 3.2**. Suppose there are three entity types, in turn, three corresponding $s^e$: $<5>_a, <7>_b, <9>_c$ where $n$ in $<n>_e$ denotes the vector size of the state representation of entity type $e$. Suppose, during training it has seen a maximum of $N^{train} := [2, 2, 2]$. During inference, suppose it observes $N^{inference} := [2, 5, 4]$. Then, as described in **Alg. 2**, $\Delta_t^{inf} := [0, 3, 2]$, meaning entity type $b$ and $c$ requires 3 and 2 observation vectors dropped out. In this case, the pre stochastically dropped out vector would include $2\times<5>_a$, $5\times<7>_b$, $4\times<9>_c$, while the post stochastically dropped out vector would include $2\times<5>_a$, $2\times<7>_b$, $2\times<9>_c$. This example shows how our approach can accommodate *any* number of entity types, each with *differing* observation sizes.
>
>
>
> ### **[Q2] Different Line Type for Our Method Visualization**
>
>
>
> Thank you for your suggestion. We have updated **Fig. 5** for improved clarity in the revised manuscript.
>
>
>
> ### **Reference**
> [1] Srivastava et al. “Dropout: A Simple Way to Prevent Neural Networks from Overfitting.” JMLR 2014.
>
> [2] Iqbal et al. “Randomized Entity-wise Factorization for Multi-Agent Reinforcement Learning.” ICML 2021.
>
> [3] Shao et al. “Complementary Attention for Multi-Agent Reinforcement Learning.” ICML 2023.

---

> > ### Comment · Reviewer_wmnB · 2024-11-26
> >
> > Thank you for the response. I will maintain my score.

---

> > > ### Author Response · Authors · 2024-11-26
> > > **Thank you**
> > >
> > > Thank you for your valuable contribution to our community. We are grateful for your positive view of our paper. We hope that your concerns have been alleviated.
> > >
> > > Best,
> > > Authors

---

### Official Review · Reviewer_v8MZ · 2024-11-04

**Soundness:** 2
**Presentation:** 3
**Contribution:** 2
**Rating:** 6
**Confidence:** 5

**Summary:**

This paper introduces FlickerFusion, a method designed to improve how multi-agent reinforcement learning (MARL) models generalize to environments with a dynamically changing number of agents. The approach involves randomly omitting agents’ observations during training, which helps the model perform better in new, dynamically changing environments in a zero-shot manner. The authors also present a modified version of Multi Particle Environments that supports variable numbers of agents to test their method.

**Strengths:**

- This paper tackles an important generalization capability in MARL.
- FlickerFusion is straightforward and can be easily integrated into various MARL frameworks.

**Weaknesses:**

- MARL is generally challenging to stabilize during training. By altering observations at each timestep, this method should increase instability in principle, yet the paper does not explain how FlickerFusion manages to achieve less variability in results compared to the baseline QMIX, as shown in Table 1 and Figure 5.
- The evaluation relies solely on the Multi Particle Environments (MPE), which has a relatively simple observational setup. It remains uncertain if FlickerFusion would be effective in more complex environments such as SMAC.
- Figures 3 and 5 in the paper would be more informative if they included quantiles instead of only the means.

**Questions:**

- In MPE, instead of randomly omitting entities, wouldn’t it make more sense to omit those farther from the ego agent based on their distance?
- How does omitting observations differ from the practice of dropping out nodes in standard neural network training?

---

> ### Author Response · Authors · 2024-11-19
> **Response to Reviewer v8MZ (Part I)**
>
> Thank you for your thoughtful feedback and valuable contribution to our research community. We gladly address your specific concerns **[C]** and questions **[Q]** in detail below.
>
>
>
> ### **[C1] How Does FlickerFusion Improve Stability?**
>
>
>
> The design of FlickerFusion allows us to *minimize the information loss* from the dropouts. This is achieved by allowing FlickerFusion to decide **“How Many to Drop?” (Sec. 3.2)** and **“Which to Drop?” (Sec. 3.2, 3.3)** on-the-fly *if* it detects OOD. In essence, **“How Many to Drop?” (Sec. 3.2)** is the domain-aware part—where  it only drops the *minimum amount required* to emulate being in-domain (see **Eq. (6)** and **Alg. 2 line 6**). It is also trained such that it learns to perform well under random entity dropouts (**Alg. 1 line 6**). I.e, as mentioned in-text: “The training is done similarly to “prepare” for the entity dropouts that the agents will experience at test time.” This is how stability problems do not arise in FlickerFusion.
>
>
>
> We further detail why this works. As detailed in **Sec. 3.1**, the root cause of this sharp rise in instability is being forced to initialize more parameters, from $\theta_Q^{D_I}$ to $\theta_Q^{D_O} := \theta_Q^{D_I} \oplus \theta_Q'$ to accommodate an increased observation vector (**page 4**). This occurs under both MLP and Attention backbones. As described in **Sec. 3.2** all prior works attempt to inject inductive biases within the newly initialized parameters, $\theta_Q'$. Orthogonally, our approach stochastically drops out observations at each timestep. We empirically show that this approach dramatically improves the trade-off of “test-time new parameters” vs. “stochastic observation dropout” (see **Tab. 4**, **Fig. 5**).
>
>
>
> ### **[C2] Environment Complexity**
>
>
>
> First, in **(1)**, we elaborate on why, despite our MPEv2 being a seemingly simple extension of MPE, the evaluation becomes much more complex and how our methodology can transfer to different environments. In **(2)**, we show a previous work [1] (ICLR 2020) that similarly evaluated their methodology on an MPE extension. Lastly, in **(3)**, we discuss how extending our problem setting to a SMAC-based environment is non-trivial, and thus, better left to a stand-alone work.
>
>
>
> **(1)** Typical MARL experiments with MPE converge in about *1 million* steps. In stark contrast, our experimental results show that all methodologies, when evaluated on MPEv2, with the addition of dynamic entity combinations and intra-trajectory entrances, do not perfectly converge even with *3 million* steps (see **Fig. 5 top-right**). This indicates that MPEv2’s dynamics are far more complex than MPE. We provide the full specifications of MPEv2 in **App. B**; MPEv2 adds significantly more challenges than MPE, including stochastic spawning, stochastic intra-trajectory entrance, out-of-domain entity compositions, and varying reward functions (some partly dense, some partly sparse) across the 6 environments and 12 benchmarks.
>
> Moreover, our FlickerFusion would transfer well to other MARL simulators, as our method is a *universal plug-in* method to any backbone (MLP or Attention-based); see **Fig. 2**. Assuming that one has the intra-trajectory OOD variant of the MARL simulators, owing to its simplicity, we believe that our FlickerFusion can be easily paired with any existing architecture and method.
>
>
>
> **(2)** We would like to direct you to consider a previous ICLR paper [1], which utilizes MPE to demonstrate the effectiveness of their approach ([https://sites.google.com/view/epciclr2020](https://sites.google.com/view/epciclr2020)). We note that both MPE and SMAC are abstractions (simplifications) of real-world scenarios, allowing researchers to *efficiently* improve their methods in a *tractable* setting. We believe that both MPE and SMAC play an integral and *complementary* role in advancing the state-of-the-art.
>
>
>
> **(3)** As mentioned by Reviewer iX1W and imDX, our open-source MPEv2 with github README documentation, is in-and-of-itself a non-trivial contribution. Evaluating intra-trajectory dynamic entity domain generalization has not yet been offered to the community. This took us a significant amount of time to code with an easy API for researchers. Therefore, akin to SMACv2 [4] and SMAC-Exp [5], extending such a setting to SMAC deserves its own stand-alone paper.
>
>
>
> ### **[C3] Visualize Uncertainty in Reward Curves**
>
>
>
> Please refer to the **global response**.

---

> ### Author Response · Authors · 2024-11-19
> **Response to Reviewer v8MZ (Part II)**
>
> ### **[Q1] Omitting Farthest Entity vs. Stochastic**
>
>
>
> You bring up an interesting potential heuristic. However, we describe here why it may *not always* be a good strategy. We would like to direct your attention to the example scenario depicted in **Fig. 1 (a)**. If a new ally agent enters intra-trajectory, but the three existing agents (top-left) omit the farthest agent—here, the newly entered agent—it will not be able to find the optimal strategy: “when an ally enters, the existing allies and the new ally can enact a dispersion strategy to spread the attention of adversaries” (**page 2**). While you raise an interesting point that it is possible to further identify good heuristics that are specific to the task, we keep our method as *general* as possible for *broad impact*. Thank you for raising an interesting future direction.
>
>
>
> ### **[Q2] Dropout vs. FlickerFusion (our method)**
>
>
>
> You raise an important parallel to the well-known neuron dropout [2] method in deep learning. Dropout stochastically drops out neural network nodes to regularize the network. Contrarily, our approach is domain generalization rather than regularization (which is focused on out-of-sample but in-domain inference) [3]. However, there are some mechanical *parallels* and *differences*, which we describe below.
>
>
>
> First, the parallel. Note that FlickerFusion drops out a (small) subset of the input vector, while dropout drops a small subset of the entire neurons. But there are also some more stark differences. FlickerFusion *selectively* drops out the number of entities depending on the in-domain space (during training, and potentially inference) and out-of-domain space (during inference), as seen in **Sec. 3.2 Eq. (6)** and **Alg. 2 line 6**. This contrasts with dropout, which drops the neurons independently across all neurons, non-selectively, and only during training. Finally, in FlickerFusion, the dropped-out entities (which are greater in size than a single neuron, specifically $\vert s^e  \vert$) are stochastically chosen across the *temporal dimension* during trajectory roll-out (see **Sec. 3.3**). [2] does not consider any temporal dimensions.
>
>
>
> ### **Reference**
> [1] Long et al. "Evolutionary Population Curriculum for Scaling Multi-Agent Reinforcement Learning." ICLR 2020.
>
> [2] Srivastava et al. “Dropout: A Simple Way to Prevent Neural Networks from Overfitting.” JMLR 2014.
>
> [3] Zhou et al. “Domain Generalization: A Survey” IEEE Transactions on Pattern Analysis and Machine Intelligence 2022.
>
> [4] Ellis et al. “SMACv2: An Improved Benchmark for Cooperative Multi-Agent Reinforcement Learning” NeurIPS 2023 Dataset & Benchmark Track.
>
> [5] Kim et al. “The StarCraft Multi-Agent Exploration Challenges : Learning Multi-Stage Tasks and Environmental Factors without Precise Reward Functions” IEEE Access (2023).

---

> > ### Comment · Reviewer_v8MZ · 2024-11-27
> > **Re**
> >
> > Thank the authors for the detailed reply. I decided to raise the rating to 6.

---

> > > ### Author Response · Authors · 2024-11-27
> > > **Thank you**
> > >
> > > Thank you for sharing your time in evaluating our paper, and supporting our community.
> > >
> > > Best, Authors.

---

### Official Review · Reviewer_iX1W · 2024-11-04

**Soundness:** 3
**Presentation:** 4
**Contribution:** 3
**Rating:** 6
**Confidence:** 4

**Summary:**

This paper addresses the MARL problem that entities like teammates or adversaries are dynamically removed or added during the inference trajectory. The key idea is to dropout observation of some sampled entities at each step, denoted as flickers, and fuse these flickers to emulate a full-view across the temporal dimension. Numerical results and demos on MPEv2 demonstrate FlickerFusion’s promising performance.

**Strengths:**

- The idea of dropping out observation of some sampled entities at each step and fuse these flickers to emulate a full-view across the temporal dimension is inspiring.
- Authors extended the widely-used MARL benchmark MPE, which can be beneficial for relevant future research.
- FlickerFusion is compared with 11 baselines, providing sufficient and solid results.
- Demo videos clearly show how FlickerFusion works.

**Weaknesses:**

- “Dropout” is the core idea and concept of this paper, but it only appears at the end of Introduction and has not been clearly defined. It would be better if authors define and emphasize it in Abstract of the beginning of Introduction, and distinguish it with the dropout operation in the neural network literature.
- Preliminaries is a bit too complex, it would be better to simplify it and relate it to the previous text, including some practical examples for illustration.
- The importance of each entity is different, but the flicker operation treats all entities equally by sampling uniformly. A module that evaluates entities’ intention / importance / uncertainty, using techniques like opponent modeling[1,2], should be developed to better guide the sampling of the flicker operation.
- The environment dynamics would change dramatically when the number of entities change. It is not clear how FlickerFusion handles this problem and realizes generalization. A clear study on this problem is vital for the application on more complex scenarios.
- The learning curves should demonstrate standard deviation.
- More related work about varying agent numbers[3] and intra-trajectory teammate shift[4] should be discussed.

[1] Georgios Papoudakis, et al. Agent Modelling under Partial Observability for Deep Reinforcement Learning.

[2] Kiarash Kazari, et al. Decentralized Anomaly Detection in Cooperative Multi-Agent Reinforcement Learning.

[3] Lei Yuan, et al. Multi-agent Continual Coordination via Progressive Task Contextualization.

[4] Ziqian Zhang, et al. Fast Teammate Adaptation in the Presence of Sudden Policy Change.

**Questions:**

See Weaknesses.

And, for this dynamic entity number problem (in MPEv2), is it possible to build a mapping that maps the dimension-varying vector observation to images (the ones you used to render the demos), so that we can simply and directly utilize a CNN to handle the issue?

---

> ### Author Response · Authors · 2024-11-19
> **Response to Reviewer iX1W (Part I)**
>
> Thank you for your thoughtful comments—each of them are highly constructive. We happily address all your concerns **[C]** and questions **[Q]** below.
>
> ### **[C1] Clarification on Dropout**
>
> Please refer to the **global response**.
>
> ### **[C2] Simplify Preliminary**
>
> First, we have *streamlined* the sub-sections. Second, we have simplified parts that are not related to the core contribution. We have *removed*:
>
> -   $\mathbb{N}$ is the set of natural numbers, and $\mathbb{N}_0 := \{0\} \cup \mathbb{N}$
> -   (possibly of different dimensions)
> -   whose support is potentially a strict subset of $A$
> -   (potentially countably infinite)
> -   Entire Transition Dynamics section
>
> Then, *rephrased* unnecessarily complex statements.
>
> We have provided intuitive visual illustrations throughout **Fig. 1**, **2**, **3**, and **4**.
>
> ### **[C3] Salience-guided Dropout**
>
> Guiding the dropout with salience-values is a good potential idea, and indeed, we have internally explored this direction. First, we describe in **(1)** the logical reasoning as to why we did not include this in our method. Finally, in **(2)**, we describe why this is better left for a future work.
>
> **(1)** As attention values have been associated with saliency, numerous works have used attention values for saliency-guiding [2, 3]—including [4] that used it for a MARL method. However, we would like to direct you to **Appendix F,** where we visualize the attention matrix from the perspective of each agent (row). Consider the left (a) matrices in **Fig. 13** to **24**. As the attention vector of each agent is highly homogenous (notice that the rows are homogenous), applying dropouts based on attention scores will result in the same entity (column) being dropped out. As aligned with this intuition, preliminary experiments using attention-guiding as introduced in [4] gave poor results.
>
> **(2)** Due to the following challenges, achieving a more advanced saliency-targeted dropout under decentralized execution is better left for future work. Following the mainstream MARL paradigm of Centralized Training Decentralized Execution (CTDE) [8, 9], it is highly challenging to *decentrally distribute* the saliency-targeted dropout across the entire fleet of agents. The challenge of *avoiding over-concentrating* the dropout on a single or few entities, is amplified under decentral execution. We solve this problem through uniform stochastic dropout, as unpacked in **Sec. 3.2** and **3.3**. It may be possible to develop a decentralized learning system that distributes saliency-targeted dropouts such that the dropouts do not result in over-concentration. Nevertheless, this is likely *contingent* on the entity combination of the learning (in-domain) phase, and unclear if it will generalize well to OOD. We leave such a challenge to future works.
>
> ### **[C4] Dynamics Change under Entity Change**
>
>
>
> Indeed, task dynamics change under entity change. First, we respond in **(1)** that this is embraced in the paper—without weakening the contribution. Next, in **(2)**, we further clarify how our method is so effective under OOD inference.
>
>
>
> **(1)** While the optimal (perfect) strategy will surely change, the objective of zero-shot domain generalization is *making best* of the *challenging situation* at test-time [4, 5]. We do not claim that our method achieves perfect optimality, but we still show empirically that it results in *improved robustness* (↑ reward, ↓ uncertainty) against *all 11 baselines*, showing that our approach is promising.
>
>
>
> **(2)** As detailed in **Sec. 3.1**, the primary challenge of zero-shot domain generalization in MARL is being forced to initialize more parameters, from $\theta_Q^{D_I}$ to $\theta_Q^{D_O} := \theta_Q^{D_I} \oplus \theta_Q'$ to accommodate an increased observation vector (**page 4**). This occurs under both MLP and Attention backbones. As described in **Sec. 3.2**, all prior works attempt to inject inductive biases within the newly initialized parameters, $\theta_Q'$. Orthogonally, our approach stochastically drops out observations at each timestep.
>
>
>
> In short, there is a trade-off between “test-time new parameter initialization” and “test-time stochastic observation,” and we empirically show that fusing the stochastic dropouts is a far better trade-off than initializing new parameters under OOD inference. We achieve this by allowing FlickerFusion to decide on-the-fly **“How Many to Drop?” (Sec. 3.2.)** and **“Which to Drop?” (Sec. 3.2., 3.3)** if it detects OOD. As it is domain-aware (**Eq. 6**), it will not drop out any observations if it does not go OOD. It is also trained (with virtually no additional computational overhead) so that it learns to perform well under random entity dropouts (**Alg. 1**).

---

> > ### Author Response · Authors · 2024-11-19
> > **Response to Reviewer iX1W (Part II)**
> >
> > ### **[C5] Visualize Uncertainty in Reward Curves**
> >
> >
> >
> > Please refer to the **global response**.
> >
> >
> >
> > ### **[C6] Two Additional Related Works**
> >
> >
> >
> > We have included these two additional works in the first paragraph of **Sec. 1**. These two works are related, but have differing problem settings. [10] is focused on continual learning for MARL (while it does include generalizability as an added benefit). [11] is focused on a *highly unique*  scenario where a teammate's policy suddenly changes. They include collaborating with uncontrollable teammates, making it completely different from ours and the *11 baselines* we include. Moreover, the parameterized agents that they control do not have intra-trajectory changing entity composition.
> >
> >
> >
> > ### **[Q1] Image Observations**
> >
> >
> >
> > Following the original MPE [6] simulator and benchmark, which is one of the top two most widely accepted MARL environment [7], we do not provide image vectors as observations. Notwithstanding, it is possible to extract screenshots at each time step using the PyGame rendering package.
> >
> >
> >
> > We do not take this approach as, first, we follow the mainstream MARL paradigm Centralized Training and Decentralized Execution (CTDE) [8, 9]. Providing a global image view will lead to emulating centralized execution. Second, this will result in a dramatic rise in computational cost, making the already high-cost MARL setting less accessible. Nevertheless, this will be an interesting future direction to explore.
> >
> >
> >
> > ### **Reference**
> > [1] Srivastava et al. “Dropout: A Simple Way to Prevent Neural Networks from Overfitting.” JMLR 2014.
> >
> > [2] Liang et al. "Rich human feedback for text-to-image generation." CVPR 2024.
> >
> > [3] Gao et al. "Learning to Incorporate Texture Saliency Adaptive Attention to Image Cartoonization." ICML 2022.
> >
> > [4] Shao et al. “Complementary Attention for Multi-Agent Reinforcement Learning.” ICML 2023.
> >
> > [5] Yuan et al. "Rl-vigen: A reinforcement learning benchmark for visual generalization." NeurIPS 2024.
> >
> > [6] Lowe et al. “Multi-Agent Actor-Critic for Mixed Cooperative-Competitive Environments.” NeurIPS 2017.
> >
> > [7] Rutherford et al. “JaxMARL: Multi-Agent RL Environments and Algorithms in JAX.” AAMAS 2024.
> >
> > [8] Jakob et al. “Stabilising Experience Replay for Deep Multi-Agent Reinforcement Learning.” ICML 2017.
> >
> > [9] Rashid et al. “QMIX: Monotonic Value Function Factorisation for Deep Multi-Agent Reinforcement Learning.” ICML 2018.
> >
> > [10] Yuan et al. “Multiagent continual coordination via progressive task contextualization.” IEEE TNNLS (Journal, 2024).
> >
> > [11] Zhang et al. "Fast teammate adaptation in the presence of sudden policy change." UAI 2023.

---

> > > ### Comment · Reviewer_iX1W · 2024-11-26
> > >
> > > Thanks for the detailed reply. I have no further questions and will maintain my rating.

---

> > > > ### Author Response · Authors · 2024-11-26
> > > > **Thank you**
> > > >
> > > > Thank you for your time and effort in evaluating our paper, and supporting the community.
> > > >
> > > > Best, Authors.

---

### Author Response · Authors · 2024-11-19
**Global Response**

We thank all the reviewers for providing thoughtful reviews. We are especially encouraged to see that the reviewers recognize:
- **Salience** of the zero-shot intra-trajectory domain generalizing **problem setting** (v8MZ, wmnB, imDX)
- **Novelty** and **simplicity** of our **method** (iX1W, v8MZ, wmnB, imDX), with mentions that it was **inspiring** (iX1W, wmnB)
- **Sufficient experimental evidence** (iX1W, wmnB, imDX)
- Extension of MPE to **MPEv2**, will be **beneficial for future research** (iX1W, imDX)
- **Demo videos** that further **aid** the **understanding** of the readers (iX1W)
- **Fully open-source** codes, benchmarks, and weights (imDX)
- **Clear presentation** (iX1W, wmnB)

To avoid redundancy, we address two concerns that were brought up by more than one reviewer here. The remaining concerns and inquiries are carefully answered in each reviewer’s reply. Note, additions or changes to the revised text have been highlighted in
${\color{blue}blue}$, for reviewers, AC and PC’s convenience. Changes to the figures are not explicitly indicated with ${\color{blue}blue}$, but are disclosed in the responses.

### **[C1] Clarification on Dropout**
Reviewers iX1W and wmnB’s feedback improves the clarity of our paper. First, we updated the **Abstract** to include a brief description of our method—mentioning dropout. Second, we provide a definition, then concretize the explanation of our augmentation as dropout-based in **Sec. 1, Introduction’s Contribution** with appropriate citation [1]:

-   “FlickerFusion stochastically drops out parts of the observation space, emulating…” (**Abstract**)
-   “The augmentation is based on the widely-adopted dropout mechanism (Srivastava et al., 2014)—where neurons (including the input layer), are randomly masked out. FlickerFusion differs from Srivastava et al. (2014) as it leverages the dropout mechanism only in the input layer, such that, the model can remain in an emulated in-domain regime, even when inferenced OOD” (**Sec. 1**)

Moreover, we have guided the readers by alluding to dropout in the following parts of **Sec. 1**:
-   “...as the entity dropout can…”
-   “...for the entity dropouts that the agents will…”
-   “...Here, dropping entities from agents' observations…”
-   “...stochastic input-space drop out at…”.

### **[C2] Visualize Uncertainty in Reward Curves**

After incorporating Reviewers v8MZ and iX1W’s feedback, our results are *even more convincing*. As visualized in **Fig. 5**, no baseline’s quartile 1 to 3 meaningfully overlap with our method’s quartile 1 to 3. We have updated the reward curves (**Fig. 3 (b)** and **Fig. 5**) to include quartile 1 to 3 shading. The figures’ captions have also been updated to reflect this. We have chosen to use quartile 1 to 3 shading as the standard deviation values are already present in **Fig. 3 (c)** and **Fig. 5 (top-left)**. This provides a more *holistic evaluation*, aligned with Reviewer v8MZ’s suggestion on using quartile shading.

### **Reference**
[1] Srivastava et al. “Dropout: A Simple Way to Prevent Neural Networks from Overfitting.” JMLR 2014.

---

### Meta-Review · Area_Chair_cSK5 · 2024-12-11

**Metareview:**

This paper includes an approach for improving domain generalization in multi-agent RL (MARL), that stochastically drops parts of the observation space during training. This helps emulate or prepare for the entity dropouts (denoted as flickers) that is experienced at test time. The authors validate the method on several tasks of the multi-particle environment (and its enhancements), demonstrating improved generalization and reduced uncertainty in dynamic environments.

In the initial reviews, the submission received positive feedback for its originality and practical relevance. Reviewers highlighted the novelty of focusing on intra-trajectory generalization in MARL and praised the clarity of the method and its experimental results. Concerns were raised about the connection to standard dropout, writing, and reporting of results/analysis. The authors provided -- clarifications during the discussion phase, included inter quartile distances in the plots, and improved the writing of preliminaries and some visualizations -- addressing these concerns satisfactorily. With these edits and clarifications, two reviewers increased their overall ratings.

The AC recommends acceptance and concurs with the unanimous positive consensus of the reviewers. This work introduces a timely and impactful contribution to improve generalization capabilities of MARL policies.

**Additional Comments On Reviewer Discussion:**

In the initial reviews, the submission received positive feedback for its originality and practical relevance. Reviewers highlighted the novelty of focusing on intra-trajectory generalization in MARL and praised the clarity of the method and its experimental results. Concerns were raised about the connection to standard dropout, writing, and reporting of results/analysis. The authors provided -- clarifications during the discussion phase, included inter quartile distances in the plots, and improved the writing of preliminaries and some visualizations -- addressing these concerns satisfactorily.

Changes in scores based on clarifications and edits from the authors:
* `iX1W` maintained a 6 rating
* `v8MZ` 5 -> 6
* `wmnB` maintained a 8
* `imDX` 5 -> 6

All reviewers acknowledged the rebuttal and responded to the authors.

---

### Decision · Program_Chairs · 2025-01-22

Accept (Poster)